# Metabolic perceptrons for neural computing in biological systems

Amir Pandi[1,5], Mathilde Koch[1,5], Peter L. Voyvodic[2], Paul Soudier[1,3], Jerome Bonnet [2], Manish Kushwaha [1] & Jean-Loup Faulon [1,3,4]

Synthetic biological circuits are promising tools for developing sophisticated systems for medical, industrial, and environmental applications. So far, circuit implementations commonly rely on gene expression regulation for information processing using digital logic. Here, we present a different approach for biological computation through metabolic circuits designed by computer-aided tools, implemented in both whole-cell and cell-free systems. We first combine metabolic transducers to build an analog adder, a device that sums up the concentrations of multiple input metabolites. Next, we build a weighted adder where the contributions of the different metabolites to the sum can be adjusted. Using a computational model fitted on experimental data, we finally implement two four-input perceptrons for desired binary classification of metabolite combinations by applying model-predicted weights to the metabolic perceptron. The perceptron-mediated neural computing introduced here lays the groundwork for more advanced metabolic circuits for rapid and scalable multiplex sensing.

[1] Micalis Institute, INRA, AgroParisTech, Université Paris-Saclay, Jouy-en-Josas, France. [2] Centre de Biochimie Structurale, INSERM U1054, CNRS UMR 5048, University of Montpellier, Montpellier, France. [3] iSSB Laboratory, Génomique Métabolique, Genoscope, Institut François Jacob, CEA, CNRS, Univ Evry, Université Paris-Saclay, 91057 Evry, France. [4] SYNBIOCHEM Center, School of Chemistry, University of Manchester, Manchester, UK. [5] These authors contributed equally: Amir Pandi, Mathilde Koch. Correspondence and requests for materials should be addressed to M.K. (email: manish.kushwaha@inra.fr) or to J.-L.F. (email: jean-loup.faulon@inra.fr)

Living organisms are information-processing systems that integrate multiple input signals, perform computations on them, and trigger relevant outputs. The multidisciplinary field of synthetic biology has combined their information-processing capabilities with modular and standardized engineering approaches to design sophisticated sense-and-respond behaviors[1–3]. Due to similarities in information flow in living systems and electronic devices[4], circuit design for these behaviors has often been inspired by electronic circuitry, with substantial efforts invested in implementing logic circuits in living cells[4–6]. Furthermore, synthetic biological circuits have been used for a range of applications including biosensors for detection of pollutants[7,8] and medically relevant biomarkers[9,10], smart therapeutics[11,12], and dynamic regulation and screening in metabolic engineering[13,14].

Synthetic circuits can be implemented at different layers of biological information processing, such as: (i) the genetic layer comprising transcription[15] and translation[16], (ii) the metabolic layer comprising enzymes[17,18], and (iii) the signal transduction layer comprising small molecules and their receptors[19,20]. Most designs implemented so far have focused on the genetic layer, developing circuits that perform computations using elements such as feedback control[21], memory systems[22,23], amplifiers[24,25], toehold switches[26], or CRISPR machinery[27,28]. However, gene expression regulation is not the only way through which cells naturally perform computation. In nature, cells carry out parts of their computation through metabolism, receiving multiple signals and distributing information fluxes to metabolic, signaling, and regulatory pathways[17,29,30]. Integrating metabolism into synthetic circuit design can expand the range of input signals and communication wires used in biological circuits, while bypassing some limitations of temporal coordination of gene expression cascades[31,32].

The number of inputs processed by synthetic biological circuits has steadily increased over the years, including physical inputs like heat, light, and small molecules such as oxygen, IPTG, aTc, arabinose and others. However, most of these circuits process input signals using digital logic, which despite its ease of implementation lacks the power that analog logic can offer[1,33,34]. The power of combining digital and analog processing is exemplified by the perceptron, the basic block of artificial neural networks inspired by human neurons[35] that can, for instance, be trained on labelled input datasets to perform binary classification. After the training, the perceptron computes the weighted sum of input signals (analog computation) and makes the classification decision (digital computation) after processing it through an activation function.

Here we describe the development of complex metabolic circuitry implemented using analog logic in whole-cell and cell-free systems by means of enzymatic reactions. For circuit design, we first employ computational design tools, Retropath[36] and Sensipath[37], that use biochemical retrosynthesis to predict metabolic pathways and biosensors. We then build and model three whole-cell metabolic transducers and an analog adder to combine their outputs. Next, we transfer our metabolic circuits to a cell-free system[38,39] in order to take advantage of the higher tunability and the rapid characterization it offers[40–42], expanding our system to include multiple weighted transducers and adders. Finally, using our integrated model fitted on the cell-free metabolic circuits we build a more sophisticated device called the metabolic perceptron, which allows desired binary classification of multi-input metabolite combinations by applying model-predicted weights on the input metabolites before analog addition, and demonstrate its utility through two examples of four-input binary classifiers. Altogether, in this work we demonstrate the potential of synthetic metabolic circuits, along with model-assisted design, to perform complex computations in biological systems.

## Results

**Whole-cell processing of multiple input metabolites.** To identify the metabolic circuits to build, we use our metabolic pathway design tools, Retropath[36] and Sensipath[37]. These tools function using a set of sink compounds at the end of a metabolic pathway, here metabolites from a dataset of detectable compounds[43], and a set of source compounds that can be used as desired inputs for the circuit. The tools then propose pathways and the enzymes that can catalyze the necessary reactions, allowing for promiscuity. Our metabolic circuit layers are organized according to the main processing functions: transduction and actuation (Fig. 1a). Transducers are the simplest metabolic circuits that function as sensing enabling metabolic pathways (SEMP)[44], consisting of one or more enzymes that transform an input metabolite into a transduced metabolite. The transduced molecule, in turn, is detected through an actuation function that is implemented using a transcriptional regulator.

We used benzoate as our transduced metabolite, its associated transcriptional activator BenR, and the responsive promoter pBen to construct the actuator layer of our whole-cell metabolic circuits[45]. To compare the shape of the response curve, we constructed the actuator layer in two formats: (i) an open-loop circuit (Fig. 1b) and (ii) a feedback-loop circuit (Supplementary Fig. 1). When compared to the open-loop format, the feedback-loop circuit has previously been shown to exhibit a linear dose–response to input[21,46]. We found that while the feedback-loop format does linearize the actuator response curve, it also reduces its dynamic range (Supplementary Fig. 1). Furthermore, the growth inhibition observed at high concentrations makes it difficult to recover the lost dynamic range by further addition of benzoate (Supplementary Fig. 6). Therefore, we selected the open-loop format due to its higher dynamic range of activation in the tested range of benzoate concentration (Fig. 1c), setting the maximum concentration of benzoate used in this work to the saturation point of this open-loop circuit.

We have previously implemented sensing-enabling metabolic pathways in whole cells for detection of molecules like cocaine, hippurate, parathion and nitroglycerin[44]. Building on that work, here we implemented three upstream transducers that convert different input metabolites into benzoate for detection by the actuator layer already tested. The transducer layers were composed of enzymes HipO for hippurate (Fig. 1d), CocE for cocaine (Fig. 1e), and an amidase coded by *vdh* gene for benzaldehyde (Fig. 1f). Compared to the benzoate output signal, we found that the transduction capacities of the three transducers were 99.6%, 49.2%, and 77.8%, respectively (Supplementary Fig. 2), indicating a partial dissipation in signal.

**A whole-cell metabolic concentration adder.** A metabolic concentration adder is an analog device composed of more than one transducer that converts their respective input metabolites into a common transduced output metabolite. For our whole-cell concentration adder, we combined two transducers to build a hippurate-benzaldehyde adder actuated by the benzoate circuit (Fig. 2a). Unlike digital bit-adders that exhibit an ON–OFF digital behavior, our metabolic adders exhibit a continuous analog behavior that is natural for metabolic signal conversion[47] (Fig. 2b and Supplementary Fig. 3). Increasing the concentration of one of the inputs at any fixed concentration of the other shows an increase in the output benzoate, and thus in the resulting fluorescence (Fig. 2b and Supplementary Fig. 3).

The maximum output signal for our analog adder, when hippurate and benzaldehyde were both at the maximum concentration of 1000 μM, was lower than the maximum signal produced by hippurate and benzaldehyde transducers alone

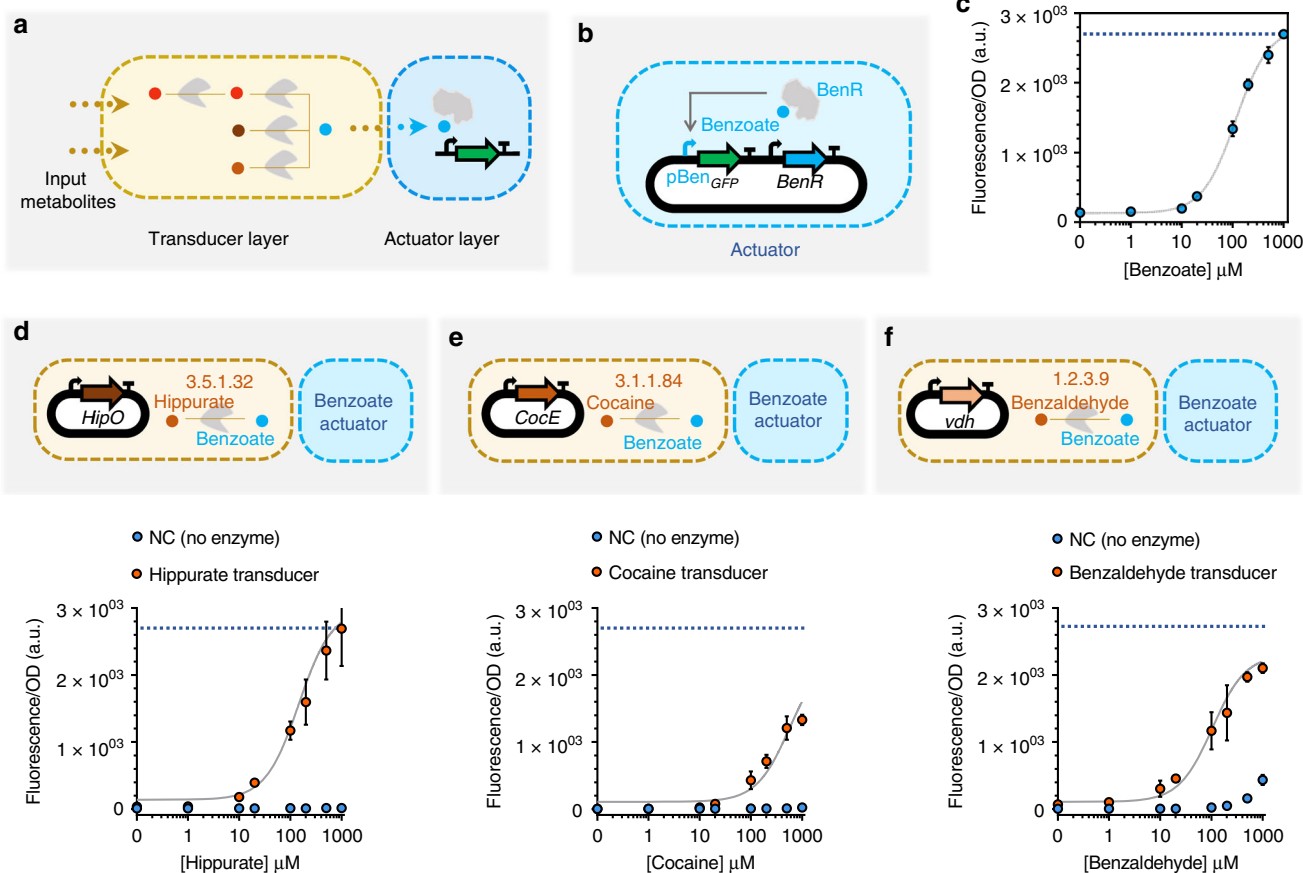

**Fig. 1** Whole-cell actuator and metabolic transducers. **a** Designed synthetic metabolic circuits using Retropath[36] or Sensipath[37] consist of a transducer layer and an actuator layer. **b** Open-loop circuit construction of the benzoate actuator, which is used downstream of transducer metabolic circuits in this work. For the open-loop circuit, the gene encoding transcription factor (TF) is expressed constitutively under control of the promoter J23101 and RBS B0032. **c** Dose-response plot of the open-loop circuit for the benzoate actuator. The gray curve is a model-fitted curve (see Methods section) for the open-loop circuit. **d**–**f** Whole-cell metabolic transducers for hippurate (**d**), cocaine (**e**) and benzaldehyde (**f**) represented in dose-response plots (orange circles) and their associated dose-response when there is no enzyme present (blue circles). The blue dotted lines refer to the maximum signal from the actuator (**c**). The transducer output, benzoate, is reported through the open-loop circuit actuator. The genes encoding the enzymes are expressed under constitutive promoter J23101 and RBS B0032. All data points and the error bars are the mean and standard deviation of normalized values from measurements taken from three different colonies on the same day. Source data are provided in the Source Data file

(Supplementary Fig. 2). However, as seen above, the difference between the maximum signal of their transducers and the actuator was smaller. The dissipation in signal could either be because of resource competition (as a result of adding more genes) or because of enzyme efficiency (as a result of poorly balanced enzyme stoichiometries). To test these two hypotheses, we investigated the effect of the enzymes on cellular resource allocation. For this purpose, the cocaine transducer and the hippurate-benzaldehyde adder were characterized by adding benzoate to these circuits (Supplementary Figs. 4 and 5). Comparing the results of these characterizations with the benzoate actuator reveals that dissipation in signal from the transducers to the actuators is due to enzyme efficiency (Supplementary Fig. 4), whereas that from the adders to the actuators is due to resource competition (Supplementary Fig. 5). The effect of the metabolic circuits on cell physiology are presented as the specific growth rate ($\mu$) of the cells harboring the circuits at different concentrations of inputs (Supplementary Figs. 6 and 7). Compared to the specific growth rate of cells containing empty plasmids ($\mu = 1.05 \pm 0.32\,\text{h}^{-1}$, the mean and standard deviation of three repeats), adding the metabolic circuits alone results only in a mild growth reduction. However, adding the metabolic circuits with their input metabolite(s) has a much

more pronounced effect on growth reduction, particularly at high concentrations.

In order to gain a quantitative understanding of the circuits' behavior, we empirically modeled their individual components to see if we were able to successfully capture their behavior. We first modeled the actuator (gray curve in Fig. 1c) using Hill formalism[48] as it is the component that is common to all of our outputs and therefore constrains the rest of our system. We then modeled our transducers, considering enzymes to be modules that convert their respective input metabolites into benzoate, which is then converted to the fluorescence output already modeled above. This simple empirical modeling strategy would be able to explain our transducer data, including the effects of enzyme efficiency, but not to account for observations made in Supplementary Fig. 5, which is why we also included resource competition is our models to explain circuits with one or more transducers. To this end, we extended the Hill model to account for resource competition following previous works[49,50], with a fixed pool of available resources for enzyme and reporter protein production that is depleted by the transducers. This extension is further presented in the Methods section. We fitted our model on all transducers, with and without resource competition (i.e., individual transducers, or transducers where another enzyme

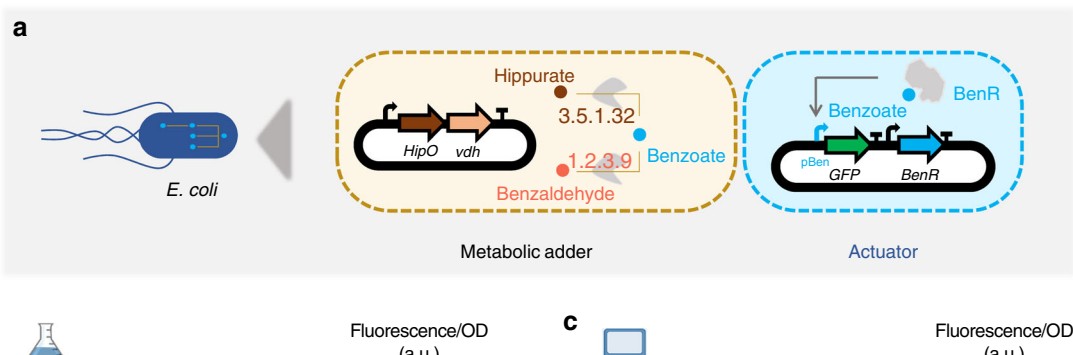

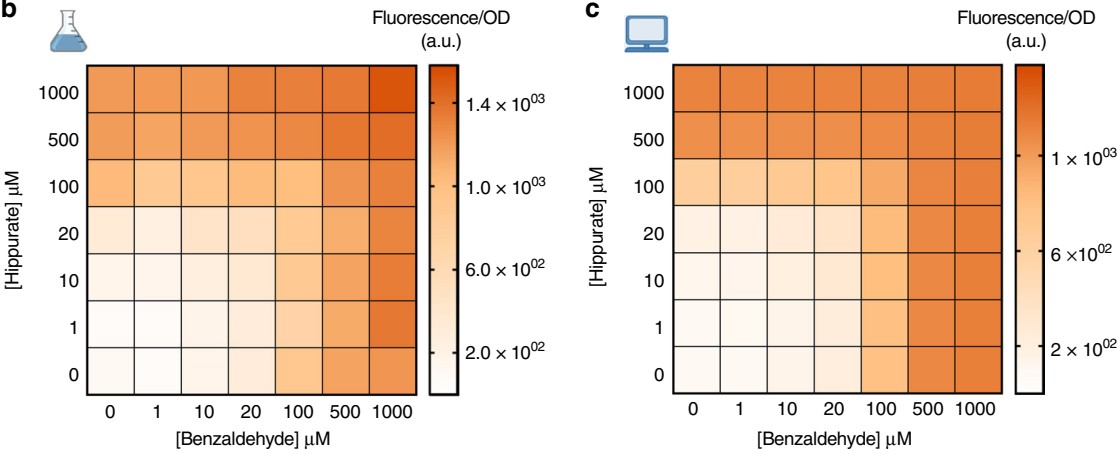

**Fig. 2** Whole-cell metabolic adder of hippurate and benzaldehyde. **a** Hippurate and benzaldehyde transducers are combined to build a metabolic adder producing a common output, benzoate, which is reported through the benzoate actuator. The genes encoding the enzymes are expressed in one operon under control of constitutive promoter J23101 and RBSs B0032 for *HipO* and B0034 for *vdh*. **b** Heatmap representing the output of the adder while increasing the concentration of both inputs, hippurate and benzaldehyde. All data points are the mean of normalized values from measurements taken from three different colonies on the same day. **c** Model simulations for experimental conditions presented in **b**. The model was fitted on transducer data and resource competition data. Source data are provided in the Source Data file

competes for the resources). This model (presented in gray lines in Figs. 1d–f and 2c), which was not trained on adder data but only on actuator, transducer, and transducers with resource competition data, recapitulates it well. This indicates that the model accounts for all important effects underlying the data. The full training process is presented in the Methods section, and a table summarizing scores of estimated goodness of fit of our model is presented in Supplementary Table 1.

**Cell-free processing of multiple metabolic inputs.** Cell-free systems have recently emerged as a promising platform[38] that provide rapid prototyping of large libraries by serving as an abiotic chassis with low susceptibility to toxicity. We took advantage of an *E. coli* cell-free system with the aim of increasing the computational potential of metabolic circuits in several ways (Fig. 3a). Firstly, a higher number of genes can be simultaneously and combinatorially used to increase the complexity and the number of inputs for our circuits. Secondly, the lower noise provided by the absence of cell growth and maintenance of cellular pathways[51] improves the predictability and accuracy of the computation. Thirdly, having genes cloned in separate plasmids enables independent tunability of circuit behavior by varying the concentration of each part individually. Finally, cell-free systems are highly adjustable for different performance parameters and components. In all, these advantages of cell-free systems enable us to develop more complex computations than the whole-cell analog adder.

Following from our recent work[52], we first characterized a cell-free benzoate actuator to be used downstream of other metabolic transducers. Figure 3a shows a schematic of the cell-free benzoate

actuator composed of a plasmid encoding the BenR transcriptional activator and a second plasmid expressing *sfGFP* reporter gene under the control of a pBen promoter. This actuator showed a higher operational range than the whole-cell counterpart (Fig. 1c). The optimal concentration of the TF plasmid (30 nM) and the reporter plasmid (100 nM) were taken from our recent study[52]. Following successful implementation of the actuator, we proceeded to build five upstream cell-free transducers for hippurate, cocaine, benzaldehyde, benzamide, and biphenyl-2,3-diol (Fig. 3c–g) that convert these compounds to benzoate. Each of the five transducers used 10 nM of enzyme DNA per reaction, except the biphenyl-2,3-diol transducer that used two metabolic enzymes with 10 nM DNA each.

Compared to its whole-cell counterpart (Fig. 1f), in the cell-free transducer reaction (Fig. 3e) benzaldehyde appears to spontaneously oxidise to benzoate without the need of the transducer enzyme vdh. This behavioral difference between the whole-cell and cell-free setups could be due to the difference in redox states inside an intact cell and the cell-free reaction mix[53,54]. Furthermore, benzamide and biphenyl-2,3-diol transducers exhibit reduction in fluorescence outputs at very high (1000 μM) input concentrations.

**Cell-free weighted transducers and adders.** After characterizing different transducers in the cell-free system that enable building a multiple-input metabolic circuit, we sought to rationally tune the transducers. Cell-free systems allow independent tuning of each plasmid by pipetting different amounts of DNA. We applied this advantage to weight the flux of enzymatic reactions in cell-free transducers (Fig. 4a). The concentration range we used was taken

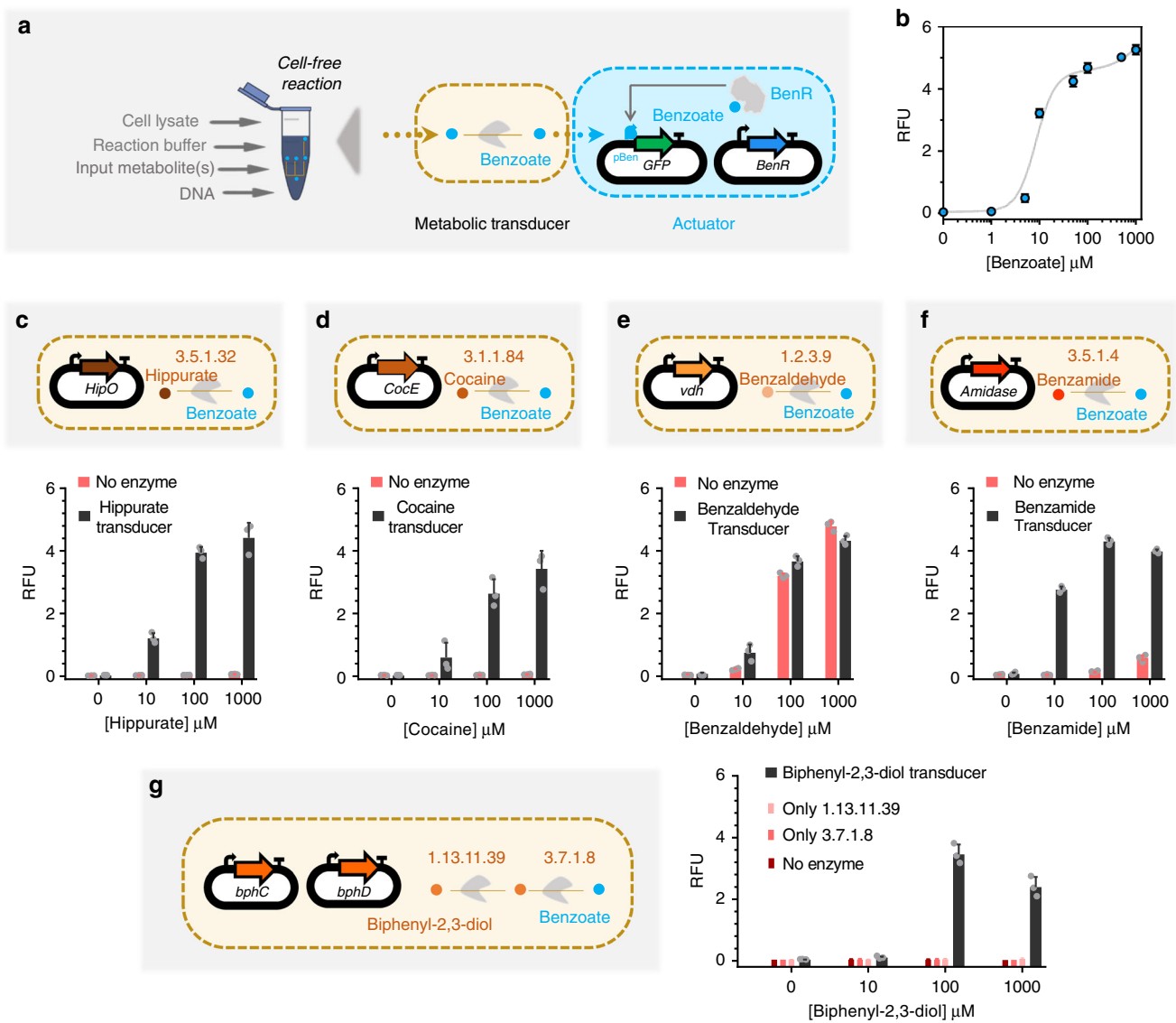

**Fig. 3** Cell-free actuator and metabolic transducers. **a** Implementing benzoate actuator and transducers in *E. coli* transcription/translation (TXTL) cell-free system. Cell-free reactions are composed of cell lysate, reaction buffer (energy source, tRNAs, amino acids, etc.) and DNA plasmids. **b** Dose–response plot of the benzoate actuator in the cell-free system with 30 nM of TF-plasmid (constitutively expressed *BenR*) and 100 nM of reporter plasmid (*pBen-sfGFP*) per reaction. The data points represent the dose–response of the actuator to different concentrations of benzoate and the gray curve is a model-fitted curve on actuator data **c**–**g**. Cell-free transducers coupled with the benzoate actuator for hippurate (**c**), cocaine (**d**), benzaldehyde (**e**), benzamide (**f**), and biphenyl-2,3-diol (**g**), which is composed of two enzymes. All enzymes are cloned in a separate plasmid under the control of a constitutive promoter J23101 and RBS B0032. 10 nM of each plasmid was added per reaction. The bars are the response of the circuits to different concentrations of input with (transducers, black bars) and without enzyme (red bars). All data are the mean and the error bars are the standard deviation of normalized values from measurements taken from three independent cell-free reactions on the same day (gray dots on each bar are individual values from which the mean and SD were calculated). (RFU relative fluorescence unit). Source data are provided in the Source Data file

from our recent study[52], in order to have an optimal expression with minimum resource competition. We built four weighted transducers for hippurate (Fig. 4b), cocaine (Fig. 4c), benza-mide (Fig. 4d) and biphenyl-2,3-diol (Fig. 4e). Increasing the concentration of the enzymes produces a higher amount of benzoate from the input metabolites, and hence higher GFP fluorescence. Compared to the others, the hippurate transducer reached higher GFP production at a given concentration of the enzyme and the input, and biphenyl-2,3-diol reached the weakest signal. For the biphenyl-2,3-diol transducer built with two enzymes (Fig. 4e), both enzymes are added at the same concentration (e.g., 1 nM of enzyme DNA indicates 1 nM each of plasmids encoding enzymes bphC and bphD). For a given

concentration of the input there is a range within which the concentration of the enzyme DNA(s) can be varied to tune the weight of the input (Supplementary Fig. 8).

Data in Fig. 4 show that similar output levels can be achieved for different input concentrations, provided the appropriate transducer concentrations are used. In the next step, we applied this finding to build hippurate-cocaine weighted adders by altering either the concentration of the enzymes or the concentration of the inputs (Fig. 5a). The fixed-input adder is an analog adder in which the concentration of inputs, hippurate and cocaine, are fixed to 100 μM and the concentration of the enzymes is altered (top panel in Fig. 5b). In this device, the weight of the reaction fluxes is continuously tunable. We then

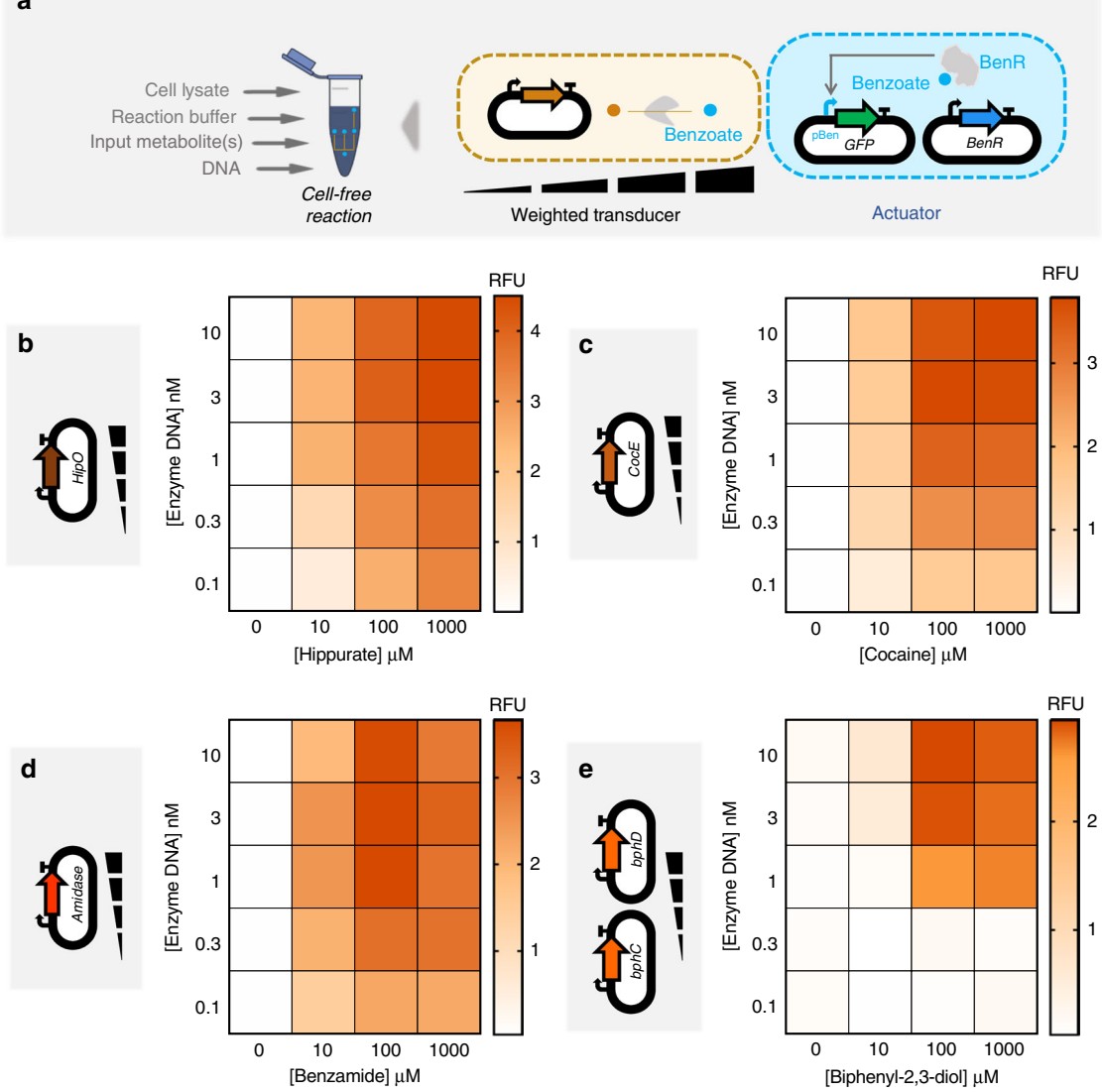

**Fig. 4** Cell-free weighted transducers characterized by varying the concentration of the enzyme DNA. **a** In the cell-free system, the circuits can be tuned by varying the amount of each enzyme pipetted per reaction. Weighted transducers are characterized by varying the concentration of the enzymes in transducers which then are reported through the benzoate actuator. The range of the concentrations was varied to get optimal expression and minimum resource competition. **b**–**e** Heatmaps representing weighted transducers at different concentrations of input molecules and enzymes DNA for hippurate (**b**), cocaine (**c**), benzamide (**d**) and biphenyl-2,3-diol (**e**). For the biphenyl-2,3-diol weighted transducer (**e**), concentrations represent those of each metabolic plasmid (e.g., 1 nM of enzyme DNA refers to 1 nM of *bphC* plus 1 nM of *bphD*). See Supplementary Fig. 9 for model results of each weighted transducer. All data are the mean of normalized values from three measurements. (RFU relative fluorescence unit). Source data are provided in the Source Data file

characterized a fixed-enzyme adder by fixing the concentration of the enzymes' DNA (1 nM for HipO, 3 nM for CocE; the cocaine signal is weaker, which is why a higher concentration of its enzyme is used) and varying the inputs, hippurate and cocaine (top panel in Fig. 5c). However, it is important to note that the observed GFP is not a direct output from the weighted adders. Instead, the adder output is transformed by the actuator to produce the GFP signal. Since the benzoate actuator has a sigmoidal response curve (Fig. 3b), the transformation by the actuator layer makes the visible output appear more switch-like (ON/OFF).

In order to have the ability to build any weighted adder with predictable results, we developed a model that accounts for the previous data. We first empirically modeled the actuator (gray curve in Fig. 3b) since all other functions are constrained by how the actuator converts metabolite data (benzoate) into a detectable

signal (GFP). We then fitted our model with individual weighted transducers (Supplementary Fig. 9) and predicted the behaviors of the weighted adders (bottom panel in Fig. 5b, c). The results shown in Fig. 5b, c indicate that our model describes the adders well, despite being fitted only on transducer data. Supplementary Table 2 summarizes the different scores to estimate the goodness of fit of our model. Briefly, the model quantitatively captures the data but tends to overestimate values at intermediate enzyme concentration ranges and does not capture the inhibitory effect observed at the high concentration of benzamide or biphenyl-2,3-diol, as this was not accounted for in the model.

Using the above strategy, we can build any weighted adder for which we have pre-calculated the weights using the model on weighted transducers. We use this ability in the following section to perform more sophisticated computation for a number of classification problems.

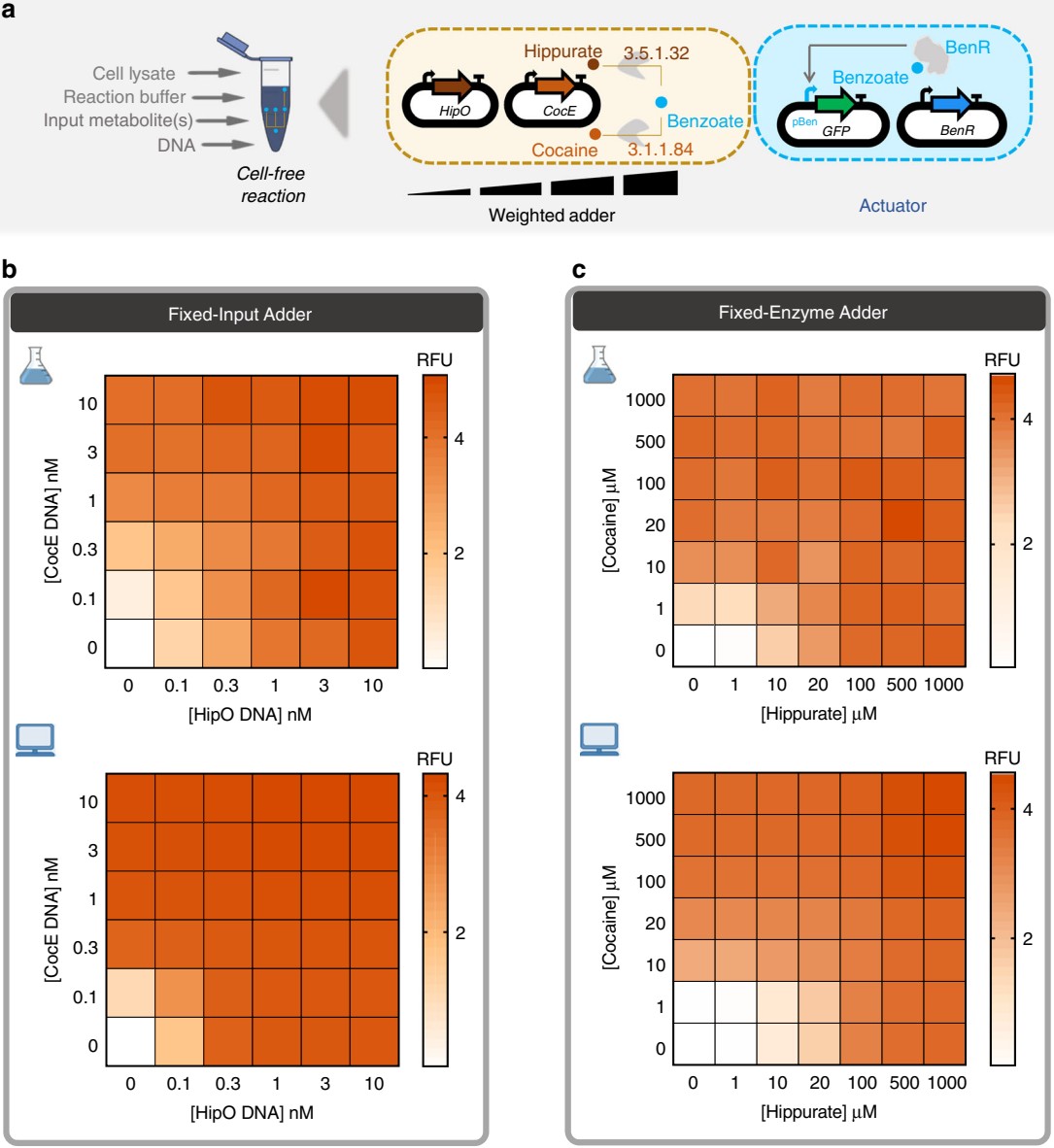

**Fig. 5** Multiple transducers are combined to shape an adder while weighting inputs or enzymes. **a** Cell-free adder characterization by varying the concentration of either inputs or enzymes producing different levels of fluorescence through the actuator. **b** Heatmap showing fixed-input adder in which the inputs, hippurate and cocaine, are fixed to 100 μM and concentrations of associated enzyme are altered by altering the concentration of plasmid DNA encoding them. Top: cell-free experiment of hippurate-cocaine fixed-input (weighted) adder. Bottom: model simulation (prediction) of hippurate-cocaine fixed-input (weighted) adder. **c** Fixed-enzyme adder with fixed concentrations of the enzymes' DNA, 1 nM for HipO and 3 nM for CocE, and various concentrations of the inputs, hippurate and cocaine. Top: Cell-free experiment of hippurate-cocaine fixed-enzyme adder. Bottom: model simulations (prediction) of hippurate-cocaine fixed-enzyme adder. All data are the mean of normalized values from three measurements. (RFU relative fluorescence unit). Source data are provided in the Source Data file

**Cell-free perceptron for binary classifications**. The perceptron algorithm was first developed to computationally mimic the neuron's ability to process information, learn, and make decisions[55]. Perceptrons are the basic blocks of artificial neural networks enabling the learning of deep patterns in datasets by training the model's input weights[56]. Like a neuron, the perceptron receives multiple input signals ($x_i$) and triggers an output depending on the weighted ($w_i$) sum of the inputs[35]. A perceptron can be used to classify a set of input combinations after it is trained on labeled data. In binary classification, the weighted sum is first calculated ($\Sigma w_i.x_i$) and an activation function ($f$), coupled with a decision threshold d, finally makes the decision: ON if f ($\Sigma w_i.x_i$) > d, OFF otherwise (Fig. 6a). The activation function can

be linear or non-linear (Sigmoid, tanh, ReLU, etc.) depending on the problem[57], although a sigmoid is generally used for classification.

Since our weighted transducer models have already been fitted on the cell-free experimental data, we checked if we could use them to calculate the weights needed to classify different combinations of two inputs: hippurate and cocaine. We tested our model on five different 2-input binary classification problems (Supplementary Fig. 10). For each problem, the two types of data were represented as a cluster of dots on the scatter plot, with the axes representing the two inputs. The fitted model was then used to identify weights needed to be applied to the weighted transducers such that a decision threshold 'd' exists to classify

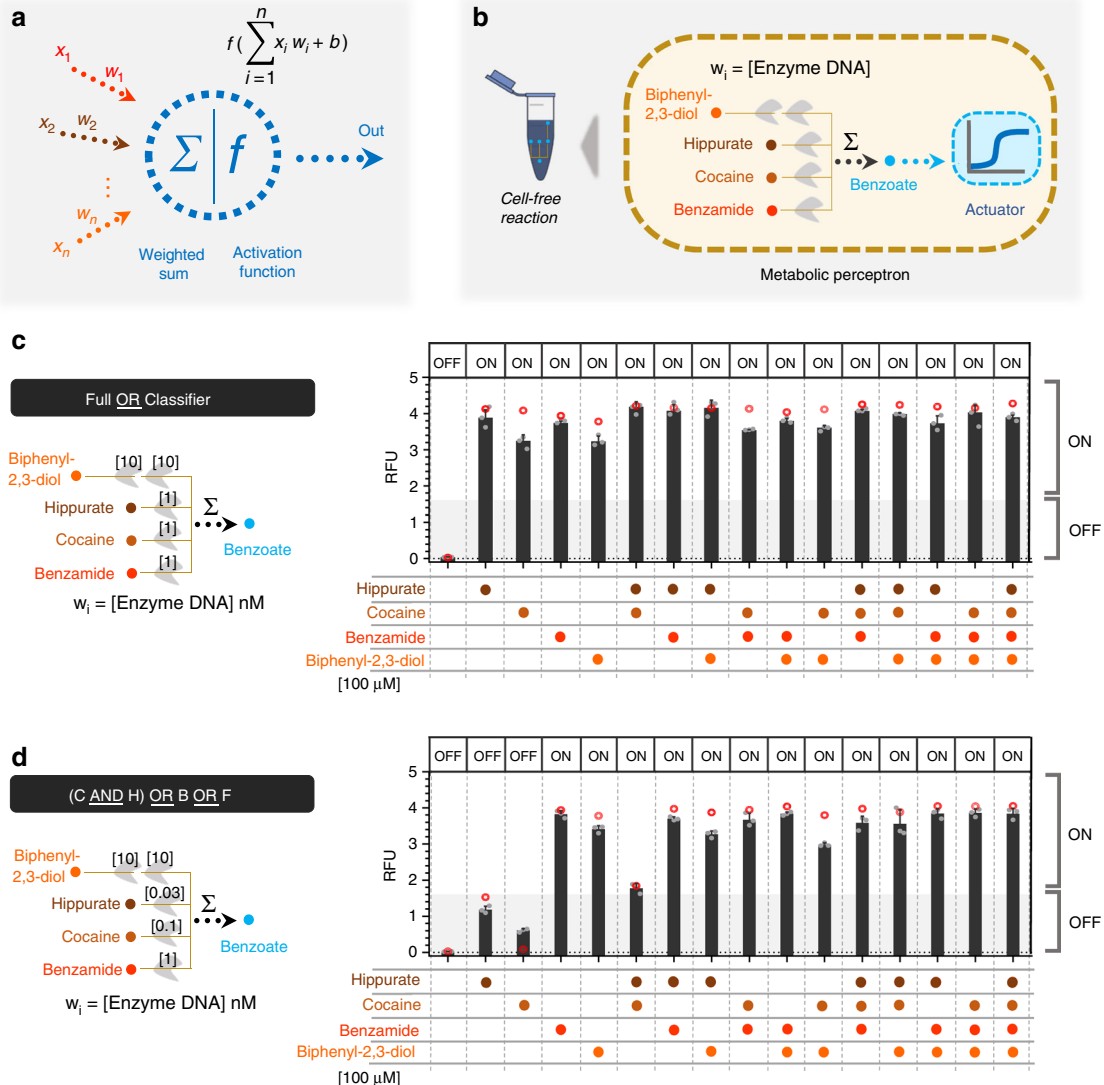

**Fig. 6** Cell-free perceptron enabling development of classifiers. **a** A perceptron scheme showing the inputs and their associated weights, the computation core, and the output. The perceptron computes the weights and actuates the weighted sum through an activation function. **b** Metabolic perceptron integrating multiple inputs and actuating an output. The benzoate actuator acts as the activation function of the perceptron reporting the sum of benzoate produced by the metabolic perceptron. Hippurate, cocaine, benzamide, and biphenyl-2,3-diol are the inputs of the metabolic perceptron fixed to 100 μM. The weights of the perceptron are the concentration of the enzymes calculated using the model made on weighted metabolic circuits (red circles). These weights are calculated to develop two classifiers using the metabolic perceptron and benzoate actuator. Full OR classifier **c**, [cocaine (C) AND hippurate (H)] OR benzamide (B) OR biphenyl-2,3-diol (F) classifier (**d**) are the two classifiers built using this metabolic perceptron. The Full OR classifier (**c**) classifies to OFF when none of the inputs is present and it passes an arbitrary threshold to ON when any of the inputs or their combinations are present. The second classifier (**d**) performs a more complex computation. The shading represents the arbitrary threshold that allows for perceptron decision making and the panel of OFF and ON at the top of the bars are the expected output of the classifiers. All data are the mean and the error bars are the standard deviation of normalized values from three measurements and red circles are the model predictions (gray dots on each bar are individual values from which the mean and SD were calculated). (RFU relative fluorescence unit). Source data are provided in the Source Data file

the two clusters into red (ON, >d) or blue (OFF, ≤d). In each binary classification, three iso-fluorescence lines threshold the data into the binary categories: ON and OFF (Supplementary Fig. 10). These theoretical classification problems demonstrate the ability of our perceptron model to successfully carry out binary classification. It is worth noting that a binary classifier whose input(s) and output are binary values can also be represented as a logic gate. Therefore, the theoretical classification functions implemented here can also be interpreted as logic gate functions. For example, the third classifier in the figure can also be represented as the equivalent logic function (H OR C) (Supplementary Fig. 10c).

Using the integrated model from our weighted transducers and adders, we next sought to design four-input binary classifiers using a metabolic perceptron, and test them experimentally. Our metabolic perceptron is a device enabling signal integration of multiple inputs with associated weights, represented by enzyme DNA concentrations (Fig. 6b). The 4-input adder performs the weighted sum and the benzoate actuator acts as the activation function of the metabolic perceptron. Similar to the 2-input binary classifications above (Supplementary Fig. 10), the weights of the four inputs can be adjusted to implement different classification functions. To illustrate the potential of building perceptrons with metabolic weighted adders, we computed adder

weights using our model for two different classifiers: a simple classifier equivalent to a full OR gate (Fig. 6c), and a more complex classifier. To define the second classifier, we used our fitted model to simulate with different weights various 4-input functions that combined AND and OR behaviors. Our simulation outcomes were most reliable for hippurate and cocaine inputs since we had previously verified our model predictions on the fixed enzyme and fixed input adders (Figs. 4 and 5). Consequently, we decided to test the classification function equivalent to a [cocaine AND hippurate] OR benzamide OR biphenyl-2,3-diol gate (Fig. 6d). Weight calculation methods are reported in the Methods section.

Finally, we used the cell-free system to implement the classifiers using the calculated weights and to execute the computations. While our perceptrons are trained in silico, they are executed in the cell-free system to predict the outcome of a given set of input signals. This is comparable to how computational perceptrons also proceed in the two phases of training and prediction. For the classifiers, the input metabolites are fixed to 100 μM, as it allows the best ON–OFF behavior for all inputs and weight-tuning according to model simulations. The model accurately predicted weights to obtain the simple full OR classifier behavior (Fig. 6d), as well as cocaine, benzamide, and biphenyl-2,3-diol weights for the second complex classifier. The initial weights computed by the model are presented in Supplementary Fig. 11. The optimal weight of HipO (hippurate transducing enzyme) was calculated to be 0.1 nM of its DNA plasmid, which leads to higher signals than predicted, particularly for the ON behavior with only hippurate. To further characterize the HipO weights at still lower concentrations of the enzyme, we performed an additional complementary characterization (Supplementary Fig. 12). Our aim here was to find a weight for HipO through which a classifier outputs a low signal (OFF) with only hippurate and high signal (ON) when coupled with other inputs. We arrived at 0.03 nM DNA for HipO enzyme which exhibited this shifting behavior between OFF and ON (Fig. 6d and Supplementary Fig. 12). Using our model-guided design and rapid cell-free prototyping on the HipO weight, we were able to design two 4-input binary classifiers. In Fig. 6c, d red circles are the weights predicted with 0.03 nM for HipO and the bars are experimental results. As noted earlier, the sigmoidal nature of the benzoate actuator's response curve (Fig. 3b) is key to achieving the OFF and ON behavior exhibited by our binary classifiers. All actual values of the model and the experiments are provided in Supplementary Table 7.

## Discussion

Computing in synthetic biological circuits has largely relied on digital logic-gate circuitry for almost two decades[5,58], treating inputs as either absent (0) or present (1). While such digital abstraction of input signals provides conceptual modularity for circuit design, it is less compatible with the physical-world input signals that vary between low and high values on a continuum[33]. As a result, digital biological circuits must carefully match input–output dynamic ranges at each layer of signal transmission to ensure successful signal processing[2,30]. More recently, the higher efficiency of analog computation on continuous input has been recognized[59], and some analog biological circuits have started emerging[21]. In this regard, using metabolic pathways for cellular computing seems like a natural progression for analog computation in biological systems[21,30].

In this study, we investigated the potential of metabolism to perform analog computations using synthetic metabolic circuits. To that end, we first established a benzoate actuator to report the output from our metabolic circuits in both whole-cell and cell-

free systems (Figs. 1c and 3b). Upstream of the actuator, we constructed hippurate, cocaine, and benzaldehyde transducers in the whole-cell system (Fig. 1d–f) and a metabolic analog adder by combining the benzaldehyde and hippurate transducers (Fig. 2). Similarly, we constructed hippurate, cocaine, benzaldehyde, benzamide, and biphenyl-2,3-diol transducers in the cell-free system (Fig. 3c–g) and weighted adders by combining them (Fig. 5). Compared to the numerous digital biological devices, which compute through multi-layered genetic logic circuits, the metabolic adder is a simple one-layered device with fast execution times.

Our computational models fitted only on the actuator and transducer data predicted adder behaviors with high accuracy (Supplementary Tables 1 and 2). This further enabled us to calculate the required weights for more complex metabolic perceptrons that compute weighted sums from multiple inputs and use them to classify the multi-input combinations in a binary manner (Fig. 6 and Supplementary Fig. 11). Although we used fixed concentrations of inputs to demonstrate the ability of our perceptrons to classify, models fitted on characterization data from weighted transducers should enable one to build classifiers for other concentrations in the operational range of the transducers (Supplementary Fig. 13). Indeed, as shown in Figs. 4 and 5, for different input concentrations in the operational range the weight of the input can be tuned through the concentration of the enzyme DNA. To the best of our knowledge, the metabolic adders and perceptrons presented in this work are the first engineered biological circuits that use metabolism for analog computation.

Unlike genetic circuits that experience expression delays[2], metabolic circuits have the advantage of faster response times since the genes have already been expressed in the system. Yet, metabolic circuits can be connected with the other layers of cellular information processing (like genetic or signal transduction layers) when needed, to build more complex sense-and-respond behaviors. The actuator layer of our perceptrons is a good example of this, where the calculated weighted sum is converted to fluorescence output via the genetic layer. In addition, we took advantage of the properties of cell-free systems, such as higher tunability and lack of toxicity[52,60], to rapidly build and characterize multiple combinations of transducer–actuator circuits. Cell-free systems can be lyophilized on paper and stored at ambient temperature for <1 year for diagnostic applications[16]. This expands the potential scope of cell-free metabolic perceptrons for use in multiplex detection of metabolic profiles in medical or environmental samples[16,52].

Here, we have built a single-layer perceptron, with positive weights, that can classify different profiles of input metabolites by applying different weights to each transducer. In the future, by adding competing or attenuating reactions that reduce the concentration of the transduced metabolite in response to an input, it may be possible to expand the training space by applying negative weights to certain inputs[61]. Furthermore, a single-layer perceptron can only classify data that is linearly separable[62], which means that it should be possible to draw a line between the two classes of data points in order for the perceptron to classify them (Supplementary Fig. 10). In contrast, multi-layer perceptrons can approximate any function[63] and can be used for more complex pattern recognition tasks[64]. With the use of bioretrosynthesis-based computational tools for metabolic pathway design, like Retropath[36] and Sensipath[37], it is theoretically possible to build multi-layer metabolic perceptrons that can classify complex patterns of metabolic states in vivo, or identify different metabolite concentrations in analytical samples (Supplementary Fig. 14). Finally, it may also be possible to apply in situ learning (within the whole-cell or cell-free environment) by applying winner selection strategies on successful classifiers[65].

However, the use of the metabolic layer for biological computing is currently underexplored. To expand the computing potential of metabolic circuits, many more metabolic parts and devices (transducers, adders, and actuators) will need to be exhaustively characterised and databases built with descriptions of activities, dynamic ranges, cross-talk, chassis dependence, cell-free composition dependence, and other functional parameters. Here, we provide a detailed method for the identification of novel parts and the step-wise building of new devices, and make our scripts available. These can form the stepping-stone for building a larger framework for fully automated design of metabolic circuits, similar to the Cello tool for automated genetic circuit design[15].

## Methods

**Designing synthetic metabolic circuits**. Retropath[36] and Sensipath[37] were used to design the metabolic circuits between potential input metabolites and detectable metabolites as outputs[43]. These tools function using a set of sink compounds, a set of source compounds, and a set of chemical rules[43,66] implementing enzyme-mediated chemical transformations. They then use retrosynthesis to propose pathways and the enzymes that can catalyze the necessary reactions, allowing promiscuity, between compounds from the sink and compounds from the source. To design the adder, the Retropath software was used with a set of detectable compounds as the sink and the molecules we wish to use as circuit inputs as the source. The results were potential pathways and the associated enzymes, which were then analyzed for feasibility. The sequences of the enzymes were codon-optimized, synthesized and implemented in E. coli or taken from a previous study.

**Molecular biology**. All plasmids were made using Golden Gate assembly in E. coli Mach1 chemically competent cells (strain W, genotype: F⁻ φ80(lacZ)ΔM15 ΔlacX74 hsdR(rₖ⁻mₖ⁺) ΔrecA1398 endA1 tonA). Whole-cell constructs were cloned in BioBrick standard vectors pSB1K3 (kanamycin resistance, pMB1 replication origin, high-copy plasmid, ~32 plasmids per genome[67]) and pSB4C5 (chloramphenicol resistance, pSC101 replication origin, low-copy plasmid, ~3.4 plasmids per genome[67]) and the genes encoding TF and all the enzymes were expressed under constitutive promoter J23101 and RBS B0032. All cell-free plasmids were cloned in pBEAST[52] (a derived vector from pBEST[68], ampicillin resistance, pMB1 replication origin, high-copy plasmid, ~32 plasmids per genome[67]). BenR cell-free plasmid and its cognate responsive promoter, pBen, expressing super-folder GFP were taken from our recent work[52]. All other cell-free enzymes were cloned under constitutive promoter J23101 and RBS B0032. Sequence and source of all the genes and parts are available in Supplementary Table 5 and the plasmids used in this study (Addgene deposit) are listed in Supplementary Table 6. Synthetic sequences were provided by Twist Bioscience. Enzymes for cloning including Q5 DNA polymerase, BsaI, and T4 DNA ligase were purchased from New England Biolabs. DNA plasmids for cell-free reactions were prepared using the Macherey-Nagel maxiprep kit.

**Characterization of whole-cell circuits**. For each circuit separate colonies of E. coli TOP10 (strain K-12, genotype: F⁻ mcrA Δ(mrr-hsdRMS-mcrBC) φ80lacZΔM15 ΔlacX74 recA1 araD139 Δ(ara-leu)7697 galU galK rpsL (Strᴿ) endA1 nupG) strains harboring the circuit plasmids were cultured overnight at 37 °C in LB with appropriate antibiotic. The next day each culture was diluted 100× in LB with antibiotics. 95 μL of fresh cultures were distributed in 96-well plate (Corning 3603) and the plate was incubated to reach the $OD_{600}$ ~ 0.1 in a plate reader (Biotek Synergy HTX). Then 5 μL of the input metabolites (100× ethanol solutions 5x diluted in LB) were added and the plate was incubated for 18 h at 37 °C. During the incubation, the $OD_{600}$ and GFP fluorescence (gain: 35, ex: 458 nm, em: 528 nm) were measured. Benzoate, hippurate, cocaine hydrochloride, benzaldehyde, benzamide and biphenyl-2,3-diol (2,3-dihydroxy-biphenyl) were purchased from Sigma-Aldrich. Permission to purchase cocaine hydrochloride was given by the French drug regulatory agency (Agence Nationale de Sécurité du Médicament et des Produits de Santé). For all chemicals, serial dilutions of 100× concentrations were prepared in ethanol. The formula presenting the results of the circuits' characterization is shown in data normalization section. The mean and standard deviation of all normalized data are provided in Supplementary Table 7.

**Cell-free extract and buffer preparation**. Cell-free E. coli extract was produced using published methods[52,69,70]. Briefly, an overnight culture of BL21 Star (DE3):: RF1-CBD₃ E. coli was used to inoculate 4 L of 2xYT-P media in six 2 L flasks at a dilution of 1:100. The cultures were grown at 37 °C with 220 rpm shaking for ~3.5–4 h until the $OD_{600}$ = 2–3. Cultures were centrifuged at 5000 × g at 4 °C for 12 min. Cell pellets were washed twice with 200 mL S30A buffer (14 mM Mg-glutamate, 60 mM K-glutamate, 50 mM Tris, pH 7.7), centrifuging after each wash at 5000 × g at 4 °C for 12 min. Cell pellets were then resuspended in 40 mL S30A buffer and transferred to pre-weighed 50 mL Falcon conical tubes where they were

centrifuged twice at 2000 × g at 4 °C for 8 and 2 min, respectively, removing the supernatant after each. Finally, the tubes were reweighed and flash frozen in liquid nitrogen before storing at −80 °C.

Cell pellets were thawed on ice and resuspended in 1 mL S30A buffer per gram of cell pellet. Cell suspensions were lysed via a single pass through a French press homogenizer (Avestin; Emulsiflex-C3) at 15,000–20,000 psi and then centrifuged at 12,000 × g at 4 °C for 30 min to separate out cellular cytoplasm. After centrifugation, the supernatant was collected and incubated at 37 °C with 220 rpm shaking for 60 min. The extract was recentrifuged at 12,000 × g at 4 °C for 30 min, and the supernatant was transferred to 12–14 kDa MWCO dialysis tubing (Spectrum Labs; Spectra/Por4) and dialyzed against 2 L of S30B buffer (14 mM Mg-glutamate, 60 mM K-glutamate, ~5 mM Tris, pH 8.2) overnight at 4 °C. The following day, the extract was re-centrifuged one final time at 12,000 × g at 4 °C for 30 min, aliquoted, and flash frozen in liquid nitrogen before storage at −80 °C.

The buffer for cell-free reactions is composed such that final reaction concentrations were as follows: 1.5 mM each amino acid except leucine, 1.25 mM leucine, 50 mM HEPES, 1.5 mM ATP and GTP, 0.9 mM CTP and UTP, 0.2 mg mL⁻¹ tRNA, 0.26 mM CoA, 0.33 mM NAD, 0.75 mM cAMP, 0.068 mM folinic acid, 1 mM spermidine, 30 mM 3-PGA, and 2% PEG-8000. Additionally, the Mg-glutamate (0–6 mM), K-glutamate (20–140 mM), and DTT (0–3 mM) levels were serially calibrated for each batch of cell-extract for maximum signal. One batch of buffer was made for each batch of extract, aliquoted, and flash frozen in liquid nitrogen before storage at −80 °C.

**Characterization of cell-free circuits**. Cell-free reactions were performed in 15.75 μL of the mixture of 33.3% cell extract, 41.7% buffer, and 25% plasmid DNA, input metabolites, and water. The reactions were prepared in PCR tubes on ice and 15 μL of each was pipetted into 384-well plates (Thermo Scientific 242764). GFP fluorescence out of each circuit was recorded in the plate reader at 30 °C (gain: 50, ex: 458 nm, em: 528 nm). The background (cell-free reaction without any plasmid) corrected fluorescence data were normalized by 20 ng μL⁻¹ of a plasmid expressing strong constitutive sfGFP (under OR2-OR1-Pr promoter[52]) and were plotted after 8 h incubation. The mean and standard deviation of all normalized data are provided in Supplementary Table 7.

**Data normalization**. For whole-cell data, we use the following normalization:

$$\text{Fluorescence(input)} = \frac{\text{GFP(input)} - \text{GFP(LB)}}{\text{OD(input)} - \text{OD(LB)}} - \frac{\text{GFP(empty\_plasmid)} - \text{GFP(LB)}}{\text{OD(empty\_plasmid)} - \text{OD(LB)}}$$

Reference: cells harboring empty plasmids

For cell-free data, we consider relative fluorescence unit (RFU):

$$\text{RFU(input)} = \frac{\text{GFP(input)} - \text{GFP(extract)}}{\text{GFP(reference)} - \text{GFP(extract)}}$$

Reference: 20 ng μL⁻¹ of a plasmid expressing the constitutive sfGFP under OR2-OR1-Pr promoter[52].

**Simulation tools and parameter fitting**. All data analysis and simulations were run on R (version 3.2.3). Dose–response curves were fitted using ordinary least squares errors and the R optim function (from Package stats version 3.2.3, using the L-BFGS-B method implementing the Limited-memory Broyden Fletcher Goldfarb Shanno algorithm, which is a quasi-Newton method). For the random parameter sampling around the mean fit, values were sampled from within ±1.96 standard error of the mean of the parameter estimation. The seed was set so as to ensure reproducibility. All simulations were run in the Rstudio development environment.

All parameters are presented in Supplementary Tables 3 and 4.

**Whole-cell model**. The whole-cell model is composed of three parts: the actuator, the transducers (which all obey the same law) and the resource competition.

$$\text{Actuator(total)} = \left( \frac{(\text{total})^{\text{hill\_a}}}{(K_M)^{\text{hill\_a}} + (\text{total})^{\text{hill\_a}}} \times fc + 1 \right) \times \text{basal}$$

where total is the concentration of the considered input (in μM), $K_M$ is the concentration that allows for half-maximum induction (in μM), also termed $IC_{50}$, hill_a is the Hill coefficient that characterizes the cooperativity of the induction system, fc is the dynamic range (in AU) and basal is the basal GFP fluorescence without input (benzoate).

$$\text{Transducer(input)} = \text{input} \times \text{range\_enz}$$

Where input is the input concentration in μM and range_enz is a dimensionless number characterizing the capacity of the enzyme to transduce the signal. When combining transducers with the actuator, transducer results are added before being fed into the actuator equation, just as benzoate concentrations are added before being converted to a fluorescent signal in the cell.

To account for resource competition, given our experimental results where there is little competition with one enzyme and significant competition with two, we used an equation including cooperativity of resource competition. This reduces

the fold change of the actuator as there are less resources available for producing transcription factors and GFP.

$$\text{Result(out)} = \text{range}_{\text{res}} \times \text{out} \times \left( \frac{(E)^{\text{nr}}}{(E)^{\text{nr}} + (coce + benz + ratio * hipo)^{\text{nr}}} \right)$$

where out is the result of the actuator transfer function before accounting for resource competition, range_res, E, nr characterize the Hill function that accounts for competition, *coce, benz and hipo* are the enzyme plasmid concentrations. *ratio* accounts for the differences in burden from different enzymes, its value around 0.8 is close to the ratio between enzyme lengths (1500 for benzaldehyde transducing enzyme and 1200 for HipO).

**Cell-free model**. The model is composed of two parts: the actuator and the transducers.

$$\text{Actuator(total)} = \left( \frac{(\text{total})^{\text{hill\_a}}}{(K_{\text{M}})^{\text{hill\_a}} + (\text{total})^{\text{hill\_a}}} \times fc + 1 \right) \times basal + lin \times 0.0001 \times total$$

where *total* is the concentration of the considered input metabolite (in μM), $K_{\text{M}}$ is the concentration that allows for half-maximum induction (in μM), also termed IC$_{50}$, hill_a is the Hill coefficient that characterizes the cooperativity of the induction system, *fc* is the dynamic range (in AU) and *basal* is the basal GFP fluorescence without input (benzoate). Lin accounts for the linearity observed in the actuator behavior at concentrations saturating the Hill transfer function.

$$\text{Transducer(input)} = \text{range}_{\text{enzyme}} \times \left( \frac{(E)^{n_E}}{(K_E)^{n_E} + (E)^{n_E}} \right) \times \left( \frac{(\text{input})^{n_{\text{input}}}}{(K_I)^{n_{\text{input}}} + (\text{input})^{n_{\text{input}}}} \right)$$

Where range_enzyme is a dimensionless number characterizing the capacity of the enzyme to transduce the signal. The activity of the enzyme is characterized by a Hill function as increasing concentrations do not lead to a linear increase but enzymes saturate ($E$ is the enzyme quantity in nM, $K_E$ and $n_E$ are its Hill constants), and similarly, input is the input metabolite concentration in μM with $K_I$ and $n\_input$ as its Hill constants.

When combining transducers, transducer results are added before being fed into the actuator equation, just as benzoate concentrations are added before being converted to the fluorescent signal in the cell.

**Model parameters fitting process**. Our fitting process is detailed in the Readme files supporting our modeling scripts provided in GitHub and is summarized here. It is done in the two steps presented here: first fitting of the actuator then fitting of the transducers.

As the first step, the actuator transfer function model (benzoate transformed into fluorescence) is fitted 100 times on the actuator data (Figs. 1c and 3b), with all actuator parameters allowed to vary. The mean, standard deviation, standard error of the mean and confidence interval were saved at 95% of the estimation of those parameters. For transducer fitting (all transducers in cell-free and all except cocaine in whole-cell, data from Fig. 1d, f, resource competition from Figs. 2b, c and 4b–e), we constrained the actuator characteristics in the following way: upper and lower allowed values are within the 95% confidence interval (or plus or minus one standard deviation from the mean for fold change and baseline in cell-free as it allowed a wider range, accounting for the decrease in actuator signal in transducer experiments without affecting the shape of the sigmoid). The initial values for the fitting process were sampled from a Gaussian distribution centered on the mean parameter estimation and spread with a standard deviation equal to the standard error of this parameter estimation. We then allowed fitting of all transducer parameters freely and of the actuator parameters within their 95% confidence interval.

Once this is done, all common parameters (actuator transfer function and resource competition) were sampled using the same procedure and fitting on the cocaine transducer was performed. To show that parameters are well constrained (proving they minimally explain the data from Fig. 1e), Supplementary Figs. 15 and 16 show results of sampling parameters from the final parameters distribution (without fitting at that stage) and how they compare to the data.

**Objective functions and model scoring**. In order to evaluate and compare our models, we used the following functions.

$$\text{RMSD} = \sqrt{\frac{\sum_1^n \left( y_i^{\text{true}} - y_i^{\text{pred}} \right)^2}{n}}$$

It measures how close the model is to the experiments. It allows for comparison of different models on the same data, the one with the smaller RMSD being better, but does not allow comparison between experiments.

$$R^2 = 1 - \frac{\sum_1^n \left( y_i^{\text{true}} - y_i^{\text{pred}} \right)^2}{\sum_1^n \left( y_i^{\text{true}} - y_{\text{mean}}^{\text{true}} \right)^2}$$

$R^2$ allows measuring the goodness of fit. When the prediction is only around the sample mean, $R^2 = 0$. When the predictions are close to the real experimental value, $R^2$ gets closer to 1, whereas it can have important negative values when the model is really far off.

$$\text{Weighted } R^2 = 1 - \frac{\sum_1^n \frac{\left( y_i^{\text{true}} - y_i^{\text{pred}} \right)^2}{\text{std}_i^2}}{\sum_1^n \frac{\left( y_i^{\text{true}} - y_{\text{mean}}^{\text{true}} \right)^2}{\text{std}_i^2}}.$$

It is a variant of $R^2$ that weights samples according to their experimental error, giving more weight or more certain samples. It otherwise has the same properties as $R^2$.

$$\text{Error percentage} = \text{abs} \left( \frac{y_i^{\text{true}} - y_i^{\text{pred}}}{y_i^{\text{true}}} \right) \times 100$$

This measures the percentage of error for each point. We present the average on all experiments in Supplementary Tables 1 and 2.

**Perceptron weights calculation**. In order to calculate the weights for the classifiers presented in Fig. 6, we followed the following procedure. First, we defined the expected results (expressed in OFFs and ONs). We also defined a list of weights to test for each enzyme (here, between 0.1 and 10 nM, as tested in our weighted transducers). Then, for each combination of enzyme weights, we simulated the outcome of the classifiers for all possible input combinations using our previously fitted model. We then tested various possible thresholds and kept the enzyme combinations for which a threshold exists that allows for the expected behavior. As the last step, we manually analyzed the classifier to keep the ones both a high difference between ON and OFF, and a minimal enzyme weight to prevent resource competitions issues that could arise as we are adding more genes than previous experiments. In order to perform clusterings presented in Supplementary Fig. 11, we sampled values uniformly within the stated ranges ([0, 2 μM] for low values and [80, 100 μM] for high values). We then simulated the results to assess the robustness of our designs. The best set of weights from this procedure to achieve the desired classification function (the 'trained' weights) are then used for the cell-free implementation.

The difference between our metabolic perceptron and an in silico perceptron is that the latter exhibits a perfect activation behavior: digital (0/1), sigmoidal, ReLU, or another activation function; its weights can be tuned exactly as desired. In our implementation of the cell-free metabolic circuits, many biological details complicate the relationship between the inputs and the activator output. We therefore used more detailed step-wise empirical modeling to account for the biology in our system rather than an off-the-shelf perceptron code that would be unable to capture all the subtleties in our data.

**Binary clustering experiments**. In order to perform the binary/2D clustering experiments, we sampled values uniformly within the stated ranges ([0, 2 μM] for low values and [80, 100 μM] for high values). For different weight (HipO and CocE) values, we simulated the fluorescence output of each of those cocaine–hippurate combinations. Moreover, for different threshold values (3, 3.5 and 4, as presented in Supplementary Fig. 10), we numerically solved for the benzoate concentration such that

$$\text{transfer(benzoate)} = \text{fluorescence\_threshold}$$

and then for values of cocaine and hippurate such that

$$\text{transducer(cocaine)} + \text{transducer(hippurate)} = \text{benzoate}$$

This equation with two unknowns gives us a curve of cocaine and hippurate values that would lie on our decided threshold for this set of weights. All combinations on the top right of that curve will be classified to ON and all combinations below will be classified as OFF.

**Biological and chemical identifiers**. In order to allow easier parsing of our article by bioinformatics tools, we provide here the identifiers of our biological sequences and chemical compounds.

Benzoate (Benzoic acid): InChI=1S/C7H6O2/c8-7(9)6-4-2-1-3-5-6/h1-5H, (H,8,9)

Hippurate (Hippuric acid): InChI=1S/C9H9NO3/c11-8(12)6-10-9(13)7-4-2-1-3-5-7/h1-5H,6H2,(H,10,13)(H,11,12)

Cocaine: InChI=1S/C17H21NO4/c1-18-12-8-9-13(18)15(17(20)21-2)14(10-12)22-16(19)11-6-4-3-5-7-11/h3-7,12-15H,8-10H2,1-2H3/t12-,13+,14-,15+/m0/s1

Benzaldehyde: InChI=1S/C7H6O/c8-6-7-4-2-1-3-5-7/h1-6H

Biphenyl-2,3-diol: InChI=1S/C12H10O2/c13-11-8-4-7-10(12(11)14)9-5-2-1-3-6-9/h1-8,13-14H

Benzamide: InChI=1S/C7H7NO/c8-7(9)6-4-2-1-3-5-6/h1-5H,(H2,8,9)

BenR (Benzoate sensitive transcription factor, Pseudomonas putida) identifier: UniProtKB - Q9L7Y6

HipO (Hippurate hydrolase (EC: 3.5.1.32), Campylobacter jejuni) identifier: UniProtKB - P45493

CocE (Cocaine esterase (EC: 3.1.1.84), Rhodococcus sp.) identifier: UniProtKB - Q9L9D7

vdh (Aryl-aldehyde oxidase (EC: 1.2.3.9), Acinetobacter johnsonii SH046) identifier: UniProtKB - D0RZT4

bphC (Biphenyl-2,3-diol 1,2-dioxygenase (EC: 1.13.11.39), Pseudomonas sp.) identifier: UniProtKB - P17297

bphD (2-Hydroxy-6-oxo-6-phenylhexa-2,4-dienoate hydrolase (EC: 3.7.1.8), Pseudomonas putida) identifier: UniProtKB - Q52036

Benzamide transforming enzyme (Amidase (EC: 3.5.1.4), Rhodococcus erythropolis) identifier: UniProtKB - B4XEY3

Sequence and source of all the genes and parts are available in Supplementary Table 5 and the plasmids used in this study (Addgene deposit) are listed in Supplementary Table 6 available at (https://www.addgene.org/browse/article/28203589/ and https://www.addgene.org/browse/article/28196338/).

**Reporting summary**. Further information on research design is available in the Nature Research Reporting Summary linked to this article.

## Data availability

The source data underlying Figs. 1c-f, 2b-c, 3c-g, 4b-e, 5b-c and 6c-d and Supplementary Figures 1–9, 12, and 13 are provided as a Source Data file. New plasmids built in this study are available from Addgene. Other raw data are available from the corresponding authors upon reasonable request.

## Code availability

All scripts and data for generating results presented in this paper are available at https://github.com/brsynth/metabolic_perceptrons.

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

## Acknowledgements

We are grateful to Pau Bernado, Annika Urbanek, and Anna Morato for helpful discussions on cell-free systems and extract preparation, and to Gottfried Otting for the BL21 Star (DE3)::RF1-CBD₃ strain. A.P. is supported by INRA (French National Institute for Agricultural Research) and the idEx Paris-Saclay interdisciplinary doctoral fellowship. M.Ko. is supported by DGA (French Ministry of Defense) and Ecole Polytechnique. P.S. and J.L.F. are supported by ANR (grant number ANR-18-CE33-0015). J.L.F. acknowledges support from BBSRC/EPSRC (grant number BB/M017702/1).

## Author contributions

A.P., M.Ko., M.Ku. and J.L.F. designed the project. A.P. designed and cloned the constructs, and performed the whole-cell experiments. A.P., P.L.V. and J.B. designed cell-free experiment platform. A.P., P.L.V. and P.S. performed cell-free experiments. M.Ko. performed computational model simulations. All authors contributed to the manuscript write-up and approved the final manuscript.

## Additional information

**Competing interests:** The authors declare no competing interests.

