## [Peer Review File · Nature Communications]

Reviewers' Comments:

Reviewer #1:

Remarks to the Author:

The manuscript "Metabolic perceptrons for neural computing in biological systems" by Pandi and co-workers explores the use of in silico-designed metabolic circuits for biological computation in both whole cells and cell-free systems. The authors have designed and implemented an analog adder (summing up the concentration of individual metabolic inputs) and a weighted adder (where such individual contributions can be adjusted). Based on these devices, this study then goes onto building two metabolic 'perceptrons' that can handle four independent metabolite inputs. Overall, the article is well balanced, and enjoyed reading the text. The authors nicely integrate experimental and computational techniques, present an extensive body of work, and the experiments and analysis thereof were—with a few exceptions, noted below—well performed and convincingly interpreted. If revised appropriately, particularly by discussing some potential shortcomings of the applications, the paper is likely to be well received by sections of the metabolic engineering, synthetic biology and modelling communities. I have no major issues with the execution of the study, but rather a number of comments and suggestions (listed below) that the authors are invited to address while preparing a revised version of their contribution:

(1) A major limitations of SynBio devices is portability. The metabolic perceptrons described in the article seem to perform well in *Escherichia coli* (or cell-free systems derived from this host). I would like to see the authors' opinion on context-dependency: considering that the metabolic effectors exploited in this work are alien to the biochemical network of *E. coli*, could these devices be transplanted into a different host? (note that this question is likewise relevant for cell-free systems). Could be such an occurrence actually exploited for metabolic engineering by taking advantage of endogenous metabolic cues as the circuit's input?

(2) P3L38: What is meant by 'toxicity' in this context? Bacterial growth inhibition? Lack of responsiveness by the device? (this is what Fig. S1b suggests, hence 'toxicity' is not an appropriate term to refer to the phenomenon).

(3) P4L4: This sentence requires expansion/further explanation: were the transducer layers taken from previous studies?

(4) P4L6-7: Define the name of these enzymes in full, and indicate their origin (this information is currently buried in Supplementary Table S5).

(5) P5L5, passim: Enzymes are not expressed, genes encoding them are. Correct the wording throughout the text accordingly.

(6) Fig. 1: Is there any impact of implanting this circuit in *E. coli* on the cell physiology? The same question pertains to the addition of the metabolites serving as inputs (e.g. benzaldehyde, which is highly reactive and definitely toxic to the cells). Key indicators (e.g. growth rates) should be provided. Why was the *E. coli* strain described in P 19L37 used for these experiments? No genotypes are provided.

(7) Page 19: Clearly indicate which are the origins of replication in these plasmids. What is the relative copy number of the plasmids when compared to each other? How does this factor influence the circuit dynamics and output?

Reviewer #2:

Remarks to the Author:

The manuscript builds on current limitations of biocomputing circuits (e.g. combinatorial logic) to

implement new models of biological computations. Specifically, authors identify the analogue vs. digital issue as the main target and address this from the metabolic vs. transcriptional standpoint. Authors built and described an in-vivo “analogue adder” and a cell-free “perceptron”. This is a very relevant and timely discussion. Method-wise, the authors adequately combine theoretical and experimental developments in an informative fashion.

However, I had difficulties in understanding how the “adder” adds and how the “perceptron” learns. I somewhat feel the authors used such terms only as metaphors. There are very interesting results in the paper, but they get diluted in trying to fit these to the metaphors. I am not convinced that the best way to describe the results is to use a “neural computing” narrative. I also fail to see the “new approach for biological computation”. I would ask the authors to address a few fundamental comments on this respect, or build a completely new story around the results (e.g. in-silico design of cell-free metabolic circuits).

- I do not understand how the adder adds. I understand an adder as a device that performs additions. It takes several inputs and *calculate* the output. Rather, what the authors show here is an accumulator, or a capacitor. In the so-called “adder” of the manuscript, two signals (inputs) generate a given output; since such output is the same, it accumulates. There is no calculation. It might be better to use the term “signal integration” (as in <https://doi.org/10.1038/nature12148>). I think this bit of the system would be better described as an “accumulation of a signal that is generated by different inputs”.

- On the same topic (adder) I find intriguing the fact that the output signal when two inputs were present was lower than the signal produced by one signal alone. That non-linearity in the integration of the signal hides very interesting dynamics. Authors point at resource competition and enzyme efficiency as the main causes. I wonder if a formal explanation of the non-linearity, as the main feature of the “adder”, could be discussed a bit more. Specifically, a discussion on how this non-linearity would affect an in-vivo usage of the “adder”.

- When transported into a cell-free system, authors claim the “adder” could increase its “computational potential”. Since a cell-free system would reduce noise and allow for tight control over enzyme concentration, do they mean increase the robustness? An increase in the number of inputs is also mentioned – is this the computational potential i.e. increase the combinatorial logic of the integration?

- I really like the interpretation of AND functions (metabolite AND enzyme = product) as weights. According to Fig 4, these weights are quite digital i.e. the enzyme input value does not really matter as long as it is high/low. However, weights (as used in perceptrons) should be able to represent wide range of values, and not just on/off.

- Coming back to the concept of an “adder”, Figure 5 shows this circuits working as an accumulator again. Since the output seems to be limited (same-ish for both inputs or one alone) the circuit could be interpreted as a simple OR logic gate. I don’t know if it is even correct to call this a “weighted adder”.

- I have similar conceptual issues with the “perceptron”. A fundamental characteristic of a perceptron is that it must be able to learn. It can be trained. However, the manuscript describes an *implementation* of a specific solution. These implementations do not learn anything. Moreover, the solution of a perceptron is a set of weights for a given problem. However, in the manuscript the cell-free “perceptrons” are given the weights as fixed values. Altogether, I don’t think such systems could be called perceptrons – nor the title of the manuscript should include “neural computing”. The use of these terms makes it difficult to interpret the results.

- A follow up comment. Looking at Figure 6c-d, what is the difference between a “classifier” and a combinatorial logic function? It would be of interest to add a discussion on similarities between the presented method for computational design and the in-silico design of combinatorial logic circuits (Cello - <http://10.0.4.102/science.aac7341>). Are the authors presenting a similar framework but for metabolism? Such a tool would be extremely valuable, but may not be the target here.

- Why did the authors choose the combinatorial function of Figure 6D? It looks like this deserves an explanation – why this one in particular?

- The conclusion that says that (page 17, line 29) “it will be possible to build multiple layers of

metabolic perceptrons that can classify complex patterns" needs an explanation. I fail to see how this would (will?) be achieved. A brief discussion on the next step towards there would be very welcome.

- Was the in-silico perceptron weight calculation done with a perceptron? Or was it a combinatorial search? If the latter, why didn't the authors build a computational perceptron for the classification problem?

Some other minor comments:

- Page 3, line 37-38. Does toxicity not affect the open-loop construct (as it does in the feedback one)?

- Figure 1B. It would be good to use line ends in the interaction depending on the regulation (inducing, arrow; repressing, perpendicular line).

- There are a couple of "not shown". This is a personal issue, but I would recommend to avoid using them by [i] showing, [ii] omitting, or [ii] explaining in details.

- Page 6, line 8 (and others). "We trained our model". Do you mean fitted? I see in Methods the fitted procedure, and the evaluation. But I don't understand the learning bit of it. Please better describe it as I think it is important.

- Page 13, line 17. Remove "A to E"

- Page 13, line 21. "The lines shown in Supp..." I'd suggest trying not to refer to SI so directly. Maybe rephrased as "conclusion this and that (Supp Figure X)".

- Page 14, line 18. I don't seem to have Supp Table S7?

- Page 22, line 15. For readability of the Equation, I would suggest using parameter names, e.g. α = cooperativity_resources, etc.

- Could authors share sequences in other formats than plain text?

Response to the reviewers:

We thank the reviewers for their thoughtful comments. We have tried to address all their concerns, and feel that our revised manuscript is now much improved as a result. Our responses below are in blue.

Reviewers' comments:

Reviewer #1 (Remarks to the Author):

The manuscript “Metabolic perceptrons for neural computing in biological systems” by Pandi and co-workers explores the use of in silico-designed metabolic circuits for biological computation in both whole cells and cell-free systems. The authors have designed and implemented an analog adder (summing up the concentration of individual metabolic inputs) and a weighted adder (where such individual contributions can be adjusted). Based on these devices, this study then goes onto building two metabolic ‘perceptrons’ that can handle four independent metabolite inputs. Overall, the article is well balanced, and enjoyed reading the text. The authors nicely integrate experimental and computational techniques, present an extensive body of work, and the experiments and analysis thereof were—with a few exceptions, noted below—well performed and convincingly interpreted. If revised appropriately, particularly by discussing some potential shortcomings of the applications, the paper is likely to be well received by sections of the metabolic engineering, synthetic biology and modelling communities. I have no major issues with the execution of the study, but rather a number of comments and suggestions (listed below) that the authors are invited to address while preparing a revised version of their contribution:

Response:

We thank the reviewer for the encouraging review. As suggested, we have now included some limitations of our work in the Discussion section.

Action: The following edits were made:

P18L36-P19L1: “However, the use of the metabolic layer for biological computing is currently underexplored. To expand the computing potential of metabolic circuits, many more metabolic parts and devices (transducers, adders, and actuators) will need to be exhaustively characterised and databases built with descriptions of activities, dynamic ranges, cross-talk, chassis dependence, cell-free composition dependence, and other functional parameters.”

(1) A major limitations of SynBio devices is portability. The metabolic perceptrons described in the article seem to perform well in *Escherichia coli* (or cell-free systems derived from this host). I would like to see the authors’ opinion on context-dependency: considering that the metabolic effectors exploited in this work are alien to the biochemical network of *E. coli*, could these devices be transplanted into a different host? (note that this question is likewise

relevant for cell-free systems). Could be such an occurrence actually exploited for metabolic engineering by taking advantage of endogenous metabolic cues as the circuit's input?

Response:

The reviewer raises some very relevant points about portability and context-dependency. Indeed, the pursuit of cross-species/ portable devices for synthetic biology has been described elsewhere as the “holy grail” ([10.1021/acssynbio.6b00256](https://doi.org/10.1021/acssynbio.6b00256)). The computational tools our current work employs (RetroPath and Sensipath) can identify enzymes from different organisms. In fact, our metabolic transducers in this work are sourced from a range of bacterial genera, *Pseudomonas*, *Campylobacter*, *Rhodococcus*, and *Acinetobacter*, but are all functional in the *E. coli* cell/ cell-free environment. So, one could expect them to work in a different cellular/ cell-free context as well. A related tool from our lab (Selenzyme, [10.1093/bioinformatics/bty065](https://doi.org/10.1093/bioinformatics/bty065)) can be used to select the best available enzyme sequence from a specified organism of interest or a closely related one, further improving chances of success in a target host.

The reviewer suggests exploiting heterologous metabolic effectors for metabolic engineering. In fact, a similar strategy has been used by the Church lab to convert 3-hydroxypropionate (3HP) into 2-methylcitrate for monitoring the pathway of interest for metabolic engineering ([10.1073/pnas.1600375113](https://doi.org/10.1073/pnas.1600375113)). They used it to optimize process parameters to improve 3HP yield by 23-fold to 4.2 g L⁻¹. We can also envision simultaneous monitoring of more than one metabolite to consider cofactor/ intermediate requirements for metabolic engineering using a metabolic perceptron.

(2) P3L38: What is meant by ‘toxicity’ in this context? Bacterial growth inhibition? Lack of responsiveness by the device? (this is what Fig. S1b suggests, hence ‘toxicity’ is not an appropriate term to refer to the phenomenon).

Response:

We thank the reviewer for pointing out the discrepancy. We had intended to convey the reduction in the dynamic range of the response curve in the feedback-loop format of the benzoate actuator. However, unlike Daniel *et al.* ([10.1038/nature12148](https://doi.org/10.1038/nature12148)), we could not rescue the dynamic range by increasing benzoate input concentration above 1000 μM due to the associated growth inhibition (=toxicity). In previous works, the maximum concentration of benzoate used was also 1000 μM ([10.1101/400788](https://doi.org/10.1101/400788) and [10.1021/acssynbio.5b00225](https://doi.org/10.1021/acssynbio.5b00225)). We have now modified the text to clarify the issue.

Action:

The following edits were made:

P3L36-P4L3: “We found that while the feedback-loop format does linearize the actuator response curve, it also reduces its dynamic range (**Supplementary Figure S1**). Furthermore, the growth inhibition observed at high concentrations makes it difficult to recover the lost dynamic range by further addition of benzoate (**Supplementary Figure S6**). Therefore, we selected the open-loop format due to its higher dynamic range of activation in the tested range

of benzoate concentration (Figure 1c), setting the maximum concentration of benzoate used in this work to the saturation point of this open-loop circuit.”

(3) P4L4: This sentence requires expansion/further explanation: were the transducer layers taken from previous studies?

Response:

In our previous works ([10.1021/acssynbio.5b00225](https://doi.org/10.1021/acssynbio.5b00225) and [10.1038/s41467-019-09722-9](https://doi.org/10.1038/s41467-019-09722-9)), we have developed whole-cell transducers for hippurate and cocaine, among other molecules. In this work, we have developed additional transducers for benzaldehyde, benzamide, and biphenyl-2,3-diol. We have now modified the text to clarify the issue.

Action:

The following edits were made:

P4L5-8: “We have previously implemented sensing-enabling metabolic pathways in whole-cells for detection of molecules like cocaine, hippurate, parathion and nitroglycerin. Building on that work, here we implemented three upstream transducers that convert different input metabolites into benzoate for detection by the actuator layer already tested.”

(4) P4L6-7: Define the name of these enzymes in full, and indicate their origin (this information is currently buried in Supplementary Table S5).

Response:

We have now provided additional details for all the enzymes in the main text: Methods section, sub-section “Biological and chemical identifiers”.

Action:

The following edits were made:

P28L14-27: “*BenR* (*Benzoate sensitive transcription factor, Pseudomonas putida*) identifier: UniProtKB - Q9L7Y6

HipO (*Hippurate hydrolase (EC: 3.5.1.32), Campylobacter jejuni*) identifier: UniProtKB - P45493

CocE (*Cocaine esterase (EC: 3.1.1.84), Rhodococcus sp.*) identifier: UniProtKB - Q9L9D7

vdh (*Aryl-aldehyde oxidase (EC: 1.2.3.9), Acinetobacter johnsonii SH046*) identifier: UniProtKB - D0RZT4

bphC (*Biphenyl-2,3-diol 1,2-dioxygenase (EC: 1.13.11.39), Pseudomonas sp.*) identifier: UniProtKB - P17297

bphD (*2-Hydroxy-6-oxo-6-phenylhexa-2,4-dienoate hydrolase (EC: 3.7.1.8), Pseudomonas putida*) identifier: UniProtKB - Q52036

Benzamide transforming enzyme (Amidase (EC: 3.5.1.4), Rhodococcus erythropolis) identifier: UniProtKB - B4XEY3”

(5) P5L5, passim: Enzymes are not expressed, genes encoding them are. Correct the wording throughout the text accordingly.

Response:

We thank the reviewer for pointing this out. We have now corrected the error throughout the main and supplementary texts of the manuscript.

Action:

The following edits were made:

P5L4: "the gene encoding transcription factor (TF) is expressed constitutively"

P5L10-11: "The genes encoding the enzymes are expressed under constitutive promoter J23101 and RBS B0032"

P7L4: "The genes encoding the enzymes are expressed in one operon"

P9L5-6: "(constitutively expressed *BenR*)"

P18L7-8: "the genes have already been expressed in the system"

P20L21-22: "and the genes encoding TF and all the enzymes were expressed under constitutive promoter J23101 and RBS B0032"

(6) Fig. 1: Is there any impact of implanting this circuit in *E. coli* on the cell physiology? The same question pertains to the addition of the metabolites serving as inputs (e.g. benzaldehyde, which is highly reactive and definitely toxic to the cells). Key indicators (e.g. growth rates) should be provided. Why was the *E. coli* strain described in P 19L37 used for these experiments? No genotypes are provided.

Response:

Adding the metabolic circuits to the cells causes mild growth reduction. However, greater reduction in growth is seen when adding increasing concentration of input metabolites and their combinations (benzoate, hippurate, cocaine, benzaldehyde). We have now included two additional figures showing the Specific Growth Rates (h^{-1}) of cells from the relevant experiments (Supplementary Figures S6 and S7) and modified the text to explain the observations.

BL21 (DE3) Star *E. coli* strains are frequently used in cell-free systems due to their large capacity for protein production. Initially, to investigate cell-free systems we chose the BL21 (DE3) Star :: RF1-CBD₃ modified strain that has a tagged release factor 1 protein since our colleagues had well established protocols for the strain and it was shown to perform indistinguishably from the untagged version ([10.1002/anie.201108275](https://doi.org/10.1002/anie.201108275)). They are now properly thanked for the strain and their kind advice in the Acknowledgments section. We also modified the text to include the genotypes of all the cell-types used in our work.

Action:

The following figures were added: Supplementary Figure S6, Supplementary Figure S7

The following edits were also made:

P5L31-P6L4: "The dissipation in signal could either be ... Comparing the results of these characterizations with the benzoate actuator reveals that dissipation in signal from the transducers to the actuators is due to enzyme efficiency (Supplementary Figure S4), whereas that from the adders to the actuators is due to resource competition (Supplementary Figure

S5). The effect of the metabolic circuits on cell physiology are presented as the specific growth rate (μ) of the cells harboring the circuits at different concentrations of inputs (**Supplementary Figures S6 and S7**). Compared to the specific growth rate of cells containing empty plasmids ($\mu = 1.05 \pm 0.32 \text{ h}^{-1}$), adding the metabolic circuits alone results only in a mild growth reduction. However, adding the metabolic circuits with their input metabolite(s) has a much more pronounced effect on growth reduction, particularly at high concentrations.”

P19L16-18: “We are grateful to Pau Bernado, Annika Urbanek, and Anna Morato for helpful discussions on cell-free systems and extract preparation, and to Gottfried Otting for the BL21 Star (DE3)::RF1-CBD₃ strain.”

P20L16-18: “All plasmids were made using Golden Gate assembly in *E. coli* Mach1 chemically competent cells (strain W, genotype: F⁻ $\phi 80(lacZ)\Delta M15 \Delta lacX74 hsdR(r_K^- m_K^+)$ $\Delta recA1398 endA1 tonA$).”

P20L36-38: “For each circuit separate colonies of *E. coli* TOP10 (strain K-12, genotype: F⁻ $mcrA \Delta(mrr-hsdRMS-mcrBC) \phi 80 lacZ \Delta M15 \Delta lacX74 recA1 araD139 \Delta(ara-leu)7697 galU galK rpsL$ (Str^R) $endA1 nupG$) strains harboring the circuit plasmids were cultured overnight at 37 °C in LB with appropriate antibiotic.”

(7) Page 19: Clearly indicate which are the origins of replication in these plasmids. What is the relative copy number of the plasmids when compared to each other? How does this factor influence the circuit dynamics and output?

Response: We have now included additional details about our plasmids in the Methods section (sub-section Molecular biology): resistance marker, replication origin, copy number (according to [10.1186/s12934-016-0610-8](https://doi.org/10.1186/s12934-016-0610-8)).

The plasmid copy number in whole-cells can influence the circuit dynamics in multiple ways. We have studied the impact of plasmid copy number *in vivo* on a transcriptional biosensor in a previous study ([10.1002/bit.26726](https://doi.org/10.1002/bit.26726)). In that work, the copy number impacted both fold-change and sensitivity, but the effects were not always straightforward. Some other studies have also looked at those effects *in vivo* using either empirical or mechanistic models, for example: [10.1186/1471-2105-13-S4-S11](https://doi.org/10.1186/1471-2105-13-S4-S11) and [10.3390/pr7030119](https://doi.org/10.3390/pr7030119).

Since plasmid copy number influences circuit outputs in complex ways, cell-free systems can be an excellent tool for analyzing its effects. They can mimic changes in plasmid copy number, which are otherwise harder to tune and control *in vivo*, by changing the amount of DNA in the reaction. In this work, in order to characterize the cell-free benzoate actuator we varied the DNA concentrations of plasmids encoding the reporter, the transcription factor, as well as the enzyme(s), which allowed us to study and model effects that would be similar to plasmid copy number changes *in vivo*.

Action: The following edits were made:

P20L18-21: "Whole-cell constructs were cloned in BioBrick standard vectors pSB1K3 (kanamycin resistance, pMB1 replication origin, high-copy plasmid, ~32 plasmids per genome) and pSB4C5 (chloramphenicol resistance, pSC101 replication origin, low-copy plasmid, ~3.4 plasmids per genome) and the genes encoding TF and all the enzymes were expressed under constitutive promoter J23101 and RBS B0032. All cell-free plasmids were cloned in pBEAST (a derived vector from pBEST, ampicillin resistance, pMB1 replication origin, high-copy plasmid, ~32 plasmids per genome)."

Reviewer #2 (Remarks to the Author):

The manuscript builds on current limitations of biocomputing circuits (e.g. combinatorial logic) to implement new models of biological computations. Specifically, authors identify the analogue vs. digital issue as the main target and address this from the metabolic vs. transcriptional standpoint. Authors built and described an in-vivo “analogue adder” and a cell-free “perceptron”. This is a very relevant and timely discussion. Method-wise, the authors adequately combine theoretical and experimental developments in an informative fashion.

Response:

We thank the reviewer for the positive remarks.

However, I had difficulties in understanding how the “adder” adds and how the “perceptron” learns. I somewhat feel the authors used such terms only as metaphors. There are very interesting results in the paper, but they get diluted in trying to fit these to the metaphors. I am not convinced that the best way to describe the results is to use a “neural computing” narrative. I also fail to see the “new approach for biological computation”. I would ask the authors to address a few fundamental comments on this respect, or build a completely new story around the results (e.g. in-silico design of cell-free metabolic circuits).

Response:

We agree with the reviewer that, like many expressions borrowed from other disciplines, the terms ‘adder’ and ‘perceptron’ as used here are probably approximate interpretations for the molecular systems they describe. In that sense, there must be some amount of metaphor fitting in its use, as has been argued for other synthetic biology terms like ‘chassis’ and ‘orthogonality’ ([10.4161/bbug.2.1.13388](https://doi.org/10.4161/bbug.2.1.13388)). However, we consider the adoption of the perceptron terminology to be important for expanding metabolic computing towards more advanced circuits.

Regarding the term ‘adder’, we also acknowledge that it has been used in literature (including in electrical engineering) for several different devices, which may result in some confusion. Two common uses of the term ‘adder’ are: (1) A digital adder/ bit-adder that adds binary values (digital inputs: 0 or 1), of which the two common sub-types are the half-adder and the full-adder; and (2) an analog adder that sums up different non-digital inputs (analog/ continuous inputs: 0.25, 0.50, etc.). When integrating the input signals, the analog adder can treat all input signals as having equal weights (analog simple adder), or it can treat different input signals as having different weights (analog weighted adder). In this work, we use the term ‘adder’ to mean digital adder only once (P5L21: “digital bit-adders”). At all other places, the term ‘adder’ is used for the second type: the analog adder. To clarify this, we have now edited the manuscript at multiple places to explicitly state that.

In our experiments, the ‘adder’ consists of several transducer enzymes receiving different input substrates (hippurate, cocaine, benzaldehyde, benzamide, and biphenyl-2,3-diol) and

converting them to a common product (benzoate). Therefore, these enzymatic conversions are equivalent to a summation because the concentration of their common product is effectively the sum of the resulting enzymatic conversions of various inputs. The benzoate in turn is converted by an 'actuator' device, with a sigmoidal response curve (Figure 3b), into a detectable/ visible signal (GFP). Next, we tune the weights of the transducers to build a 'weighted adder', so that the output signal is dependent on the weights of the transducer enzymes. The 'weighted adder' when combined with a sigmoidal 'actuator' forms a perceptron which can be used for classification.

The significant novelty of our work is the ability to generate multiple different binary classifiers from the same transducers by changing input weights according to predictions from a fitted 'perceptron' model. To the best of our knowledge, such metabolic computation using a perceptron has not been done before. Therefore, we believe that it constitutes a "new approach for biological computation".

Action: The following edits were made:

P5L17: "A metabolic concentration adder is an **analog** device"

P5L27: "The maximum output signal for our **analog** adder"

P7L24: "more complex computations than the whole-cell **analog** adder."

P12L5: "The fixed-input adder is an **analog** adder in which"

P17L25: "and a metabolic **analog** adder by combining"

- I do not understand how the adder adds. I understand an adder as a device that performs additions. It takes several inputs and *calculate* the output. Rather, what the authors show here is an accumulator, or a capacitor. In the so-called "adder" of the manuscript, two signals (inputs) generate a given output; since such output is the same, it accumulates. There is no calculation. It might be better to use the term "signal integration" (as in <https://doi.org/10.1038/nature12148>). I think this bit of the system would be better described as an "accumulation of a signal that is generated by different inputs".

Response:

We agree with the reviewer that our analog adder takes multiple metabolites as inputs and converts them into a common metabolite (benzoate). We consider the concentration of the common metabolite, or integrated signal, as the analog sum of the input metabolites. This operation is like that of an electronic analog adder (also called analog summator or summing amplifier) that sums up the input voltages it receives. Our use of the term 'adder' is also consistent with Daniel *et al.* ([10.1038/nature12148](https://doi.org/10.1038/nature12148)), referred to by the reviewer, where the 'signal integration' terminology is used interchangeably with the terms: 'adder', 'adder circuit', and 'analog adders'.

However, we concede to the reviewer that in certain situations, our adder device can be considered an accumulator or a capacitor. An accumulator stores intermediate results of multi-step calculations and a capacitor stores electrical energy. If there is substantial time-delay between benzoate production and its conversion to the GFP output, for example due to slow or delayed actuator expression, it may then become appropriate to see our device as a 'storage'

device, i.e. an accumulator or a capacitor. However, in our case the BenR actuator is expressed concurrently with the transducer enzymes. As a result, it is already available for actuation as soon as the transducers convert the input metabolite into benzoate. Therefore, we prefer to keep the term ‘adder’ that is more consistent with our device’s function, as well as with the terminology of Daniel *et al.*

- On the same topic (adder) I find intriguing the fact that the output signal when two inputs were present was lower than the signal produced by one signal alone. That non-linearity in the integration of the signal hides very interesting dynamics. Authors point at resource competition and enzyme efficiency as the main causes. I wonder if a formal explanation of the non-linearity, as the main feature of the “adder”, could be discussed a bit more. Specifically, a discussion on how this non-linearity would affect an in-vivo usage of the “adder”.

Response:

As seen in the heat-map of the whole-cell metabolic adder in Figure 2b, and subsequently in more detail in Supplementary Figure S3, when only one input is present the output signal is lower than when both the inputs are present. We’d like to point out to the reviewer that the only situation where the output signal from two inputs was lower than the output signal from a single input (Supplementary Figure S2) was when there were two enzymes present with the two inputs (Hippurate-Benzaldehyde Adder) and only one enzyme present with the single input (Hippurate/Benzaldehyde/ Cocaine Transducer). These results are from two different systems/ devices and are not strictly comparable, but they are shown in the figure as a way of demonstrating the potential loss in the final signal and setting the stage for further investigation in Supplementary Figures S4 and S5. We find that decrease in signal from the 2-input adder is due to resource competition. When comparing comparable systems, for example when two enzymes are present for either one or two inputs, the signal for one input is indeed lower than signal from two inputs. To clarify these points, we have now simplified Supplementary Figure S2 and edited the legends for Supplementary Figures S2, S4 and S5.

Action:

The following figure was modified: Supplementary Figure S2

The following edits were made in the legends for Supplementary Figures S2, S4 and S5:

PS4L8-10: “The actuator (blue) and transducer (beige) data and error bars are from the results presented in **Figure 1**. The adder (orange) data and error bars are from the results presented in **Figure 2**.”

PS6L17-18: “The cocaine transducer (+ benzoate actuator) with benzoate input shows a behavior similar or close to the benzoate actuator **alone**.”

PS7L17-19: “The adder (+ benzoate actuator) with benzoate input shows a behavior similar to the adder (+ benzoate actuator) with hippurate and benzaldehyde **inputs**.”

- When transported into a cell-free system, authors claim the “adder” could increase its “computational potential”. Since a cell-free system would reduce noise and allow for tight

control over enzyme concentration, do they mean increase the robustness? An increase in the number of inputs is also mentioned – is this the computational potential i.e. increase the combinatorial logic of the integration?

Response:

Transporting the adder from the whole-cell to the cell-free system increases its “computational potential” in multiple ways. In the manuscript, we list four of these parameters at P7L17-23 that would be more difficult to explore *in vivo* due to cellular toxicity effects and the limited control over plasmid types and copy numbers. These include: (1) the number of different enzymes that can be expressed simultaneously in different combinations, (2) the lower noise and higher predictability of expression in cell-free systems, (3) the greater control over different enzymes’ numbers in the system, and (4) the ability to tune other expression/ performance parameters of the cell-free lysate. In particular, the ability to tune the enzyme DNA concentrations and combine them in different proportions, which we use to weight our input signals, is critical for the construction of the weighted adder used in our metabolic perceptron.

- I really like the interpretation of AND functions (metabolite AND enzyme = product) as weights. According to Fig 4, these weights are quite digital i.e. the enzyme input value does not really matter as long as it is high/low. However, weights (as used in perceptrons) should be able to represent wide range of values, and not just on/off.

Response:

We agree with the reviewer that weights in a perceptron should be able to represent a wide range of values. In Figure 4, our experiments were aimed at rationally tuning the four transducers to identify the best conditions for applying the desired weights. Since the different transducer enzymes have different catalytic properties, differences in sensitivity to DNA amounts are expected. Indeed, we found that the hippurate transducer was the most tunable and the biphenyl-2,3-diol transducer was the least. However, it is important to note that the observed GFP is not a direct output from the weighted adder. Instead, the adder output is transformed by the actuator to produce the GFP signal. Since the benzoate actuator has a sigmoidal response curve (Figure 3b), the transformation by the actuator layer makes the visible output appear more switch-like (ON / OFF). This is a very desirable feature for the rest of our experiments, notably the perceptron, where non-linearity is very important.

Since we vary only enzyme DNA concentrations as weights in our perceptrons, while keeping the input metabolite concentrations fixed, only the vertical axes of the heat-maps in Figure 4 are relevant for tuning the weights. To clarify the point, we have now edited the text in the manuscript and re-plotted the data in the new Supplementary Figure S8, where the effect of tuning input weights by varying enzyme DNA concentrations is evident. The tunable behavior in response to varying enzyme DNA concentrations is also evident in Figure 5b.

Action:

The following figure was added: Supplementary Figure S8

The following edits were made:

P10L16-18: “For a given concentration of the input there is a range within which the concentration of the enzyme DNA(s) can be varied to tune the weight of the input (**Supplementary Figure S8**).”

P12L11-15: “However, it is important to note that the observed GFP is not a direct output from the weighted adders. Instead, the adder output is transformed by the actuator to produce the GFP signal. Since the benzoate actuator has a sigmoidal response curve (**Figure 3b**), the transformation by the actuator layer makes the visible output appear more switch-like (ON / OFF).”

P15L30-32: “As noted earlier, the sigmoidal nature of the benzoate actuator’s response curve (**Figure 3b**) is key to achieving the “OFF” and “ON” behavior exhibited by our binary classifiers.”

- Coming back to the concept of an “adder”, Figure 5 shows this circuits working as an accumulator again. Since the output seems to be limited (same-ish for both inputs or one alone) the circuit could be interpreted as a simple OR logic gate. I don’t know if it is even correct to call this a “weighted adder”.

Response:

As explained in a previous response to the reviewer, we prefer the use of the term ‘adder’ for our device as it is consistent with its behavior and similar use elsewhere in literature.

Further, we agree with the reviewer that the adder results in Figure 5 can be interpreted as constituting OR gate logic. The distinction between the two interpretations lies in whether we are interested in the continuous or the discrete nature of the output. Looking at the top heat-map in Figure 5b (experimental heat map), from the analog point of view, the continuous gradient of fluorescence from the bottom-left to top-right corner represents a summation of input signals. From a digital point of view, by applying thresholds to both axes the same gradient can be discretized into four quadrants representing OR gate logic, something we do for the theoretical 2-input classifications in Supplementary Figure S10. That digital logic is sometimes described as a “special case” of analog logic agrees with this dual interpretation. Some of these concepts, as relevant for synthetic biology, were reviewed in a 2015 paper ([10.1109/TBCAS.2015.2461446](https://doi.org/10.1109/TBCAS.2015.2461446)). The conceptual equivalent of the digital OR gate logic ($A \text{ OR } B$) in set theory is called a ‘union’, represented as $(A \cup B)$. A similar idea in mathematics is called ‘logical disjunction’, represented as $(A \vee B)$ or $(A + B)$. In many programming languages, it is represented as $(A \text{ I } B)$, and in electronics as $(A + B)$. In summary, arithmetic addition and logical OR are conceptually similar. Therefore, depending on the viewpoint or discipline, they can be interpreted as one or the other.

We would call the Fixed-Input adder in Figure 5b a “weighted adder” because despite the same concentrations of input metabolites added (100 μM each of cocaine and hippurate), the adder

gives us different results based on the weights (enzyme DNA concentrations) applied to these inputs.

- I have similar conceptual issues with the “perceptron”. A fundamental characteristic of a perceptron is that it must be able to learn. It can be trained. However, the manuscript describes an *implementation* of a specific solution. These implementations do not learn anything. Moreover, the solution of a perceptron is a set of weights for a given problem. However, in the manuscript the cell-free “perceptrons” are given the weights as fixed values. Altogether, I don’t think such systems could be called perceptrons – nor the title of the manuscript should include “neural computing”. The use of these terms makes it difficult to interpret the results.

Response:

We agree with the reviewer that the solution of a perceptron is a set of weights for a given problem. As explained above in a previous response, the learning and the prediction in this work are done by an *in silico* model, whereas the solutions are implemented/ tested experimentally. For clarity, we would like to point out again the method we used to design the solutions we implemented. First, we fitted our *in silico* model using experimental data from the actuator and the weighted transducers. Next, we used our *in silico* model to select the best set of weights needed to perform the desired classification functions, just like a computational perceptron would do. Finally, we used the cell-free system to implement the classifiers using the calculated weights and to execute the computations. So, while our perceptrons are trained *in silico*, they are executed in the cell-free system to predict the outcome of a given set of input signals. Computational perceptrons also proceed in the two phases of training and prediction. However, this is often obscured by the fact that both phases are performed *in silico*, whereas our setup makes the distinction between the two phases much more obvious, one being computational and the other being experimental. Once a computational perceptron has learned weights on the data, it also stops training and uses fixed weights for the implementation/ testing. To explain this better, we have added a few sentences to the manuscript.

Action:

The following edits were made:

P15L10-14: “Finally, we used the cell-free system to implement the classifiers using the calculated weights and to execute the computations. While our perceptrons are trained *in silico*, they are executed in the cell-free system to predict the outcome of a given set of input signals. This is comparable to how computational perceptrons also proceed in the two phases of training and prediction.”

- A follow up comment. Looking at Figure 6c-d, what is the difference between a “classifier” and a combinatorial logic function?

Response:

A “classifier” is any model/ process that can sort different types of datapoints into the classes to which they belong. An example from machine learning could be a trained model that can take

an input set of pictures and classify them into 'cat', 'dog', and 'sheep'. When the number of classes is only two, for example only 'cat' and 'dog', the classifier is called a binary classifier. Any binary classifier can, upon application of a threshold, be represented as a digital logic function whose two classes are: 'ON' and 'OFF' (or '1' and '0'). Stated differently, a binary classifier whose input(s) and output are binary values can be represented as a logic gate.

In this work, we simulated five 2-input binary classifiers that can classify any 2-input datapoint into 'blue' (below threshold fluorescence, or '0') and red (above threshold fluorescence, or '1') (Supplementary Figure S10). The value of an input datapoint is a combination of cocaine and hippurate concentrations that represents its coordinates on the two axes. The classification decision is based on whether the fluorescence for the datapoint lies above or below the threshold set by the iso-fluorescence line. While we present this work as a binary classifier, the classification functions implemented can also be interpreted as logic gate functions. For example, the classifier in Supplementary Figure S10c can be represented as logic function (H OR C). Similarly, the 4-input binary classifiers implemented in Figures 6c and 6d can be represented as logic functions (C OR H OR B OR F) and ((C AND H) OR B OR F). A key feature in both sets of classifiers is that they use the same transducers, (H, C) in Supplementary Figure S10c and (C, H, B, F) in Figure 6. Yet, depending on the weights applied to these transducers, they can represent diverse classification functions.

Action:

The following edits were made:

P14L25-29: "It is worth noting that a binary classifier whose input(s) and output are binary values can also be represented as a logic gate. Therefore, the theoretical classification functions implemented here can also be interpreted as logic gate functions. For example, the third classifier in the figure can also be represented as the equivalent logic function (H OR C) (Supplementary Figure S10c)."

- It would be of interest to add a discussion on similarities between the presented method for computational design and the in-silico design of combinatorial logic circuits (Cello - <http://10.0.4.102/science.aac7341>). Are the authors presenting a similar framework but for metabolism? Such a tool would be extremely valuable, but may not be the target here.

Response:

We thank the reviewer for referencing Cello (10.1126/science.aac7341), which is an advanced *in silico* tool for automated design of genetic circuits, and by extension combinatorial genetic logic. On the contrary, our current work is an early proof-of-principle aimed at applying perceptron-mediated neural computing using metabolic circuits. While we do not provide an advanced automated tool like Cello at the moment, we do provide a detailed method and make our scripts available, which can form the stepping-stone for such an automation framework. As more data become available from transducer, actuator, and adder characterizations, it should be possible in the future to build the framework for advanced metabolic computing. We have modified the text to discuss such a possibility.

Action:

The following edits were made:

P19L1-4: “Here, we provide a detailed method for the identification of novel parts and the step-wise building of new devices, and make our scripts available. These can form the stepping-stone for building a larger framework for fully automated design of metabolic circuits, similar to the Cello tool for automated genetic circuit design.”

- Why did the authors choose the combinatorial function of Figure 6D? It looks like this deserves an explanation – why this one in particular?

Response:

Apart from implementing the “full OR” classifier, we also wanted to test the ability of our approach for designing a different classifier that combines AND and OR behaviors. We used our trained perceptron model to simulate multiple such gates, finally deciding on a classifier function for which our simulation output was the most reliable: ((C AND H) OR B OR F). This was because the part of the expression that was different from the first classifier (C AND H) had been verified previously by our model predictions (Figure 5). We have now modified the text to explain the decision.

Action:

The following edits were made:

P15L1-7: “To define the second classifier, we used our fitted model to simulate with different weights various 4-input functions that combined AND and OR behaviors. Our simulation outcomes were most reliable for hippurate and cocaine inputs since we had previously verified our model predictions on the fixed enzyme and fixed input adders (**Figures 4 and 5**). Consequently, we decided to test the classification function equivalent to a “[cocaine AND hippurate] OR benzamide OR biphenyl-2,3-diol” gate (**Figure 6d**).”

- The conclusion that says that (page 17, line 29) “it will be possible to build multiple layers of metabolic perceptrons that can classify complex patterns” needs an explanation. I fail to see how this would (will?) be achieved. A brief discussion on the next step towards there would be very welcome.

Response:

We agree with the reviewer that multi-layer metabolic perceptrons are not yet imminent in the field. However, there is no theoretical limit to building them. In order to build a single-layer perceptron, as we did in this study, we needed three things: input signals (here, input metabolites), a way of weighting them (here, transducer enzymes) and a way of performing non-linear activation (here, our transcription factor actuator layer). In our implementation, we plugged the output from this single layer into a fluorescent signal. To obtain a multi-layer perceptron, we will need to plug the output of this first layer as the input of the second layer of perceptron. While various strategies could be used to implement this, we have now presented one potential implementation of a multi-layer perceptron in the new Supplementary Figure S14.

However, any such implementation will need additional orthogonal metabolic parts to serve as transducers and actuators in the new perceptron layer.

Action:

The following figure was added: Supplementary Figure S14

The following edits were made:

P18L28-32: “With the use of bioretrosynthesis-based computational tools for metabolic pathway design, like Retropath³⁶ and Sensipath³⁷, it is theoretically possible to build multi-layer metabolic perceptrons that can classify complex patterns of metabolic states *in vivo*, or identify different metabolite concentrations in analytical samples (Supplementary Figure S14).”

- Was the in-silico perceptron weight calculation done with a perceptron? Or was it a combinatorial search? If the latter, why didn't the authors build a computational perceptron for the classification problem?

Response:

The difference between our metabolic perceptron and an *in silico* perceptron is that the latter exhibits a perfect activation behavior: digital (0 / 1), sigmoidal, ReLU, or another activation function; its weights can be tuned exactly as desired. In our implementation of the cell-free metabolic circuits: (i) each transducer has its own characteristics (Figure 4), (ii) the actuation is performed by a TF-based biosensor layer which is not perfectly sigmoidal (Figure 3b), and (iii) the range to tune the weights is limited by enzyme efficiency and resource competition at higher expression levels. We therefore used more detailed step-wise empirical modeling to account for the biology in our system rather than an off-the-shelf perceptron code that would be unable to capture all the subtleties in our data. We have now modified the methods section of the manuscript to explain this.

Action:

The following edits were made:

P26L20-26: “The difference between our metabolic perceptron and an *in silico* perceptron is that the latter exhibits a perfect activation behavior: digital (0 / 1), sigmoidal, ReLU, or another activation function; its weights can be tuned exactly as desired. In our implementation of the cell-free metabolic circuits, many biological details complicate the relationship between the inputs and the activator output. We therefore used more detailed step-wise empirical modeling to account for the biology in our system rather than an off-the-shelf perceptron code that would be unable to capture all the subtleties in our data.”

Some other minor comments:

- Pag 3, line 37-38. Does toxicity not affect the open-loop construct (as it does in the feedback one)?

Response:

Yes, toxicity (or growth inhibition) with increasing concentrations of benzoate affects both the open-loop and the feedback-loop designs of the actuator as they are carried by the same cell-type susceptible to higher concentrations of benzoate. As a result, the loss in dynamic range of the feedback-loop actuator cannot be rescued by increasing input benzoate concentrations. In previous works too, the maximum concentration of benzoate used was 1000 μM (10.1101/400788 and 10.1021/acssynbio.5b00225). We have now modified the text to clarify the issue (also mentioned in response to reviewer 1).

Action:

The following edits were made:

P3L36-P4L3: “We found that while the feedback-loop format does linearize the actuator response curve, it also reduces its dynamic range (**Supplementary Figure S1**). Furthermore, the growth inhibition observed at high concentrations makes it difficult to recover the lost dynamic range by further addition of benzoate (**Supplementary Figure S6**). Therefore, we selected the open-loop format due to its higher dynamic range of activation in the tested range of benzoate concentration (**Figure 1c**), setting the maximum concentration of benzoate used in this work to the saturation point of this open-loop circuit.”

- Figure 1B. It would be good to use line ends in the interaction depending on the regulation (inducing, arrow; repressing, perpendicular line).

Response:

We thank the reviewer for pointing this out. We have now modified all the relevant figures.

Action:

Modifications were made to the following figures: 1b, 2a, 3a, 4a, 5a, and S1a.

- There are a couple of “not shown”. This is a personal issue, but I would recommend to avoid using them by [i] showing, [ii] omitting, or [iii] explaining in details.

Response:

We thank the reviewer for pointing this out. The first occurrence of the “not shown” has been explained and the second one has been omitted.

Action:

The following edits were made:

P6L12-16: “This simple empirical modeling strategy would be able to explain our transducer data, including the effects of enzyme efficiency, but not account for observations made in **Supplementary Figure S5**, which is why we also included resource competition in our models to explain circuits with one or more transducers.”

P15L14-16: “For the classifiers, the input metabolites are fixed to 100 μM , as it allows the best ON-OFF behavior for all inputs and weight-tuning according to model simulations—(~~results not shown~~).”

- Page 6, line 8 (and others). “We trained our model”. Do you mean fitted? I see in Methods the fitted procedure, and the evaluation. But I don’t understand the learning bit of it. Please better describe it as I think it is important.

Response:

We thank the reviewer for pointing this out. The training indeed applies only to the perceptron weights calculations, the rest of the model was fitted. We have now modified the wording accordingly throughout the text. The “Full model training process” is now called “Model parameters fitting process”, where we have modified the text for better explanation and included references to the figures whose experimental data was used for the fitting steps. Further, we have clarified what we call trained weights in the “Perceptron weights calculation” section.

Action:

The following edits were made:

P1L30: “computational model fitted on experimental data”

P3L7: “our integrated model fitted on the”

P6L19: “We fitted our model on all transducers”

P12L21: “We then fitted our model”

P12L24: “despite being fitted only on transducer data”

P14L14 “already been fitted on the cell-free”

P14L19: “The fitted model was then used”

P17L33: “Our computational models fitted only on”

P17L39: “models fitted on characterization data”

P24L24: “Model parameters fitting process”

P24L25-27: “Our fitting process is detailed in the Readme files supporting our modeling scripts provided in GitHub and is summarized here. It is done in the two steps presented here: first fitting of the actuator then fitting of the transducers.”

P24L30: “on the actuator data (Figures 1c and 3b),”

P24L33-34: “For transducer fitting (all transducers in cell-free and all except cocaine in whole-cell, data from Figures 1d and 1f, resource competition from Figures 2b and 2c , 4b, 4c, 4d, 4e),”

P25L9-10: “To show that parameters are well constrained (proving they minimally explain the data from Figure 1e)”

P26L7-9: “Then, for each combination of enzyme weights, we simulated the outcome of the classifiers for all possible input combinations using our previously fitted model.”

P26L17-18: “The best set of weights from this procedure to achieve the desired classification function (the ‘trained’ weights) are then used for the cell-free implementation.”

- Page 13, line 17. Remove “A to E”

Action:

We removed “A to E” in the revised manuscript.

P14L16-17: “We tested our model on five different 2-input binary classification problems (Supplementary Figure S10).”

- Page 13, line 21. “The lines shown in Supp...” I’d suggest trying not to refer to SI so directly. Maybe rephrased as “conclusion this and that (Supp Figure X)”.

Action:

The following edits were made:

P14L21-23: “In each binary classification, three iso-fluorescence lines threshold the data into the binary categories: ON and OFF (**Supplementary Figure S10**).”

- Page 14, line 18. I don’t seem to have Supp Table S7?

Response:

Supplementary Tables S6 and S7 were uploaded as Excel files during the original submission.

Action:

Supplementary Tables S6 and S7 of the original submission have now been combined into a single file and will be uploaded as the ‘Source Data’ file. The data have also been provided in Supplementary Table S7.

- Page 22, line 15. For readability of the Equation, I would suggest using parameter names, e.g. α = cooperativity_resources, etc.

Action:

According to the reviewer’s suggestion we have now simplified and shortened our parameters names throughout the text.

- Could authors share sequences in other formats than plain text?

Action:

We have deposited all our plasmids and their plasmid maps (sequences) with Addgene and listed the details in Supplementary Table S6: name, description, and Addgene ID.

Reviewers' Comments:

Reviewer #1:

Remarks to the Author:

I have read with pleasure the revised version of the ms by Pandi and collaborators, in which all my concerns and comments raised during the first round of revision have been addressed. I endorse the publication of this article in Nature Communications, and congratulate the authors on such a fine piece of work.

Reviewer #2:

Remarks to the Author:

The authors have made many improvements to the manuscript, and my comments were addressed very carefully. I am very satisfied with their response to reviewers. So I think the community will benefit from reading it.

I'm still not convinced we can call this device a 'perceptron'... On this issue, I like the paragraph the authors included talking about a 'metabolic perceptron' vs. an 'in silico perceptron'. I guess that the full name ('metabolic perceptron') finds its place.

Response to the reviewers:

Our responses to the reviewers below are in blue.

REVIEWERS' COMMENTS:

Reviewer #1 (Remarks to the Author):

I have read with pleasure the revised version of the ms by Pandi and collaborators, in which all my concerns and comments raised during the first round of revision have been addressed. I endorse the publication of this article in Nature Communications, and congratulate the authors on such a fine piece of work.

We thank the reviewer for the endorsement and the congratulatory remarks.

Reviewer #2 (Remarks to the Author):

The authors have made many improvements to the manuscript, and my comments were addressed very carefully. I am very satisfied with their response to reviewers. So I think the community will benefit from reading it.

We are pleased to hear that the reviewer found our responses very satisfactory, and considers our work of benefit to the community.

I'm still not convinced we can call this device a 'perceptron'... On this issue, I like the paragraph the authors included talking about a 'metabolic perceptron' vs. an 'in silico perceptron'. I guess that the full name ('metabolic perceptron') finds its place.

We are glad that the reviewer likes the text we added to compare our metabolic perceptron to an *in silico* perceptron. We share with them the hope that the 'metabolic perceptron' will find its appropriate place in the field.